# Morph-specific selection drives phenotypic divergence in color polymorphic tawny owls (*Strix aluco*) in Northern Europe

Arianna Passarotto [1,2,3,6] ✉, Moritz David Lürig [1,4,6], Esa Aaltonen[5] & Patrik Karell[1]

There is a long tradition in using genetically based color polymorphisms in natural populations to study evolutionary processes. Despite growing evidence for continuous phenotypic variation within discrete morphs, we still know little about how this shapes selective dynamics. Here, using 43 years of plumage color data from a Finnish tawny owl population (*Strix aluco*), we show that gray and brown morphs exhibit substantial intra-morph variation, which has diverged over time. Plumage in the brown morph became increasingly pigmented, while the gray morph showed an abrupt shift toward lighter coloration. By examining both adult and offspring plumage, we identified morph-specific drivers of these trends: in gray owls, reduced pigmentation appears linked to extreme winters that eroded standing genetic variation, likely constraining their evolutionary response. In contrast, brown morph dynamics were shaped by an interaction between plumage coloration, reproductive success, and breeding timing, along with stronger temperature effects during the pre-fledging period. These findings suggest that intra-morph variation determines each morph's response to selection pressures, potentially influencing their ability to track shifting phenotypic optima. Our work highlights the relevance of phenotypic variation within genetically discrete morphs for evolutionary processes, including how populations respond to environmental change.

Color variation in animal populations spanning spatially or temporally variable environments is a classic research theme in ecology and evolutionary biology[1]. Consistent intraspecific variation in color often stems from color polymorphisms (hereafter CPs), discrete color variants, whose expression results from direct genetic variation rather than from environmental influence[2,3]. Based on these assumptions, CPs have long been used as visual markers of genetic variation, thereby providing a simple but effective tool to study a variety of evolutionary processes such as natural and sexual selection, and speciation in a range of different organisms[4–6]. However, alongside discrete variation, many systems also exhibit continuous variation within each morph[7–10], which may reflect the action of more complex regulatory mechanisms that operate downstream of the main gene controlling the polymorphism[11,12], or may arise from environmental sensitivity and phenotypic plasticity[13,14]. Regardless of the underlying mechanism, the presence of intra-morph continuous color variation might complicate our

understanding of the evolutionary processes that generate and maintain CPs, as it can be expected that discrete vs continuous color traits may be under different evolutionary dynamics[12,14,15].

The persistence of CPs within populations implies a balance of selective forces that keep the average fitness of each morph relatively stable over time, so that no morph can outcompete the other[3,16]. Mechanisms that lead to equivalent average fitness among color morphs include negative frequency-dependent selection[17,18] and variation in selection across environmental gradients (Fig. 1A)[19,20]. Shifts in such gradients - whether driven by climate, land use, invasive species, or other factors - can reshape the fitness landscape of polymorphic populations, which can have potentially dramatic consequences for the persistence of the polymorphism[4,21]: in a fully discrete system, changes to the selective regime would only affect the frequency of the morphs in a population[15], whereas in species where the morphs additionally show continuous color variation, within-morph variation may act

¹Evolutionary Ecology Unit, Department of Biology, Lund University, Lund, Sweden. ²Universidad de Sevilla, Seville, Spain. ³School of Biodiversity, One Health and Veterinary Medicine, University of Glasgow, Glasgow, Scotland, UK. ⁴Florida Museum of Natural History, University of Florida, Gainesville, FL, USA. ⁵Independent researcher, Lohja, Finland. ⁶These authors contributed equally: Arianna Passarotto, Moritz David Lürig. ✉e-mail: ariannapassarotto84@gmail.com

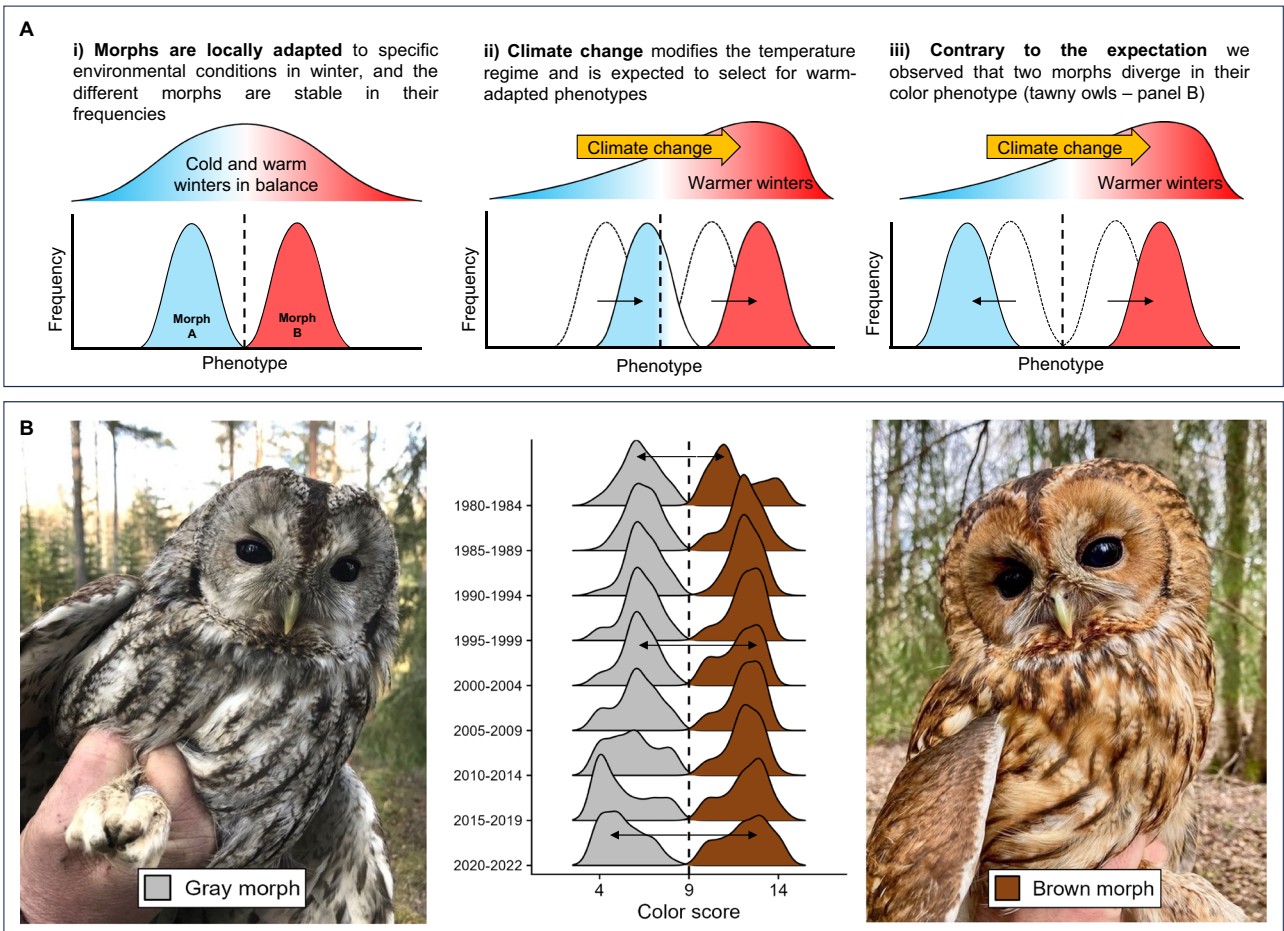

**Fig. 1 | Color polymorphisms under a changing environment. A** Schematic illustration of a color polymorphism with continuous variation among morphs adapted to different temperatures, Morph A to cold and Morph B to warm temperatures. Panel (i) shows stable phenotype frequencies around each morph's optimal color phenotype. The vertical dashed line indicates the physiological limit for color expression due to genetic and developmental constraints. As climate change induces a shift towards higher temperatures (ii), selection favors warmer-tolerant phenotypes. Despite this, Morph A cannot surpass its physiological limit to the production of coloration, leading to a decline in its frequency under warmer conditions. Conversely, Morph B not only increases in frequency but also exhibits a higher occurrence of warm-tolerant color phenotypes within the morph. In the present study (iii), we observed a pattern that does not match this expectation: instead of moving in the same direction along an axis of coloration, the two morphs of tawny owls (*Strix aluco*) diverge in their coloration. **B** Density plot of plumage color scores from 4 (pale gray) to 14 (dark brown) recorded in the Finnish population between 1978 and 2022. The vertical dashed line marks the cut-off used to assign the two morphs in the population over 40 years (i.e., the putative physiological limit of color expression). While initially both morphs appeared to move toward a more brown-dominated coloration (higher color scores), since the early 2000s the morphs have started to diverge. Photos by Chiara Morosinotto (gray morph) and Gian Luigi Bucciolini (brown morph).

as fuel for directional selection. Intraspecific variation would therefore enable the morphs to trace new phenotypic optima[15] (Fig. 1A), however, only up to the physiological limits - e.g., for color expression set by genetic and developmental constraints in pigment production[22–24]. Therefore, novel or changing selection pressures might not only affect morph frequencies directly, but also through the distribution of color phenotypes within a morph.

Here, using a 43-year dataset of plumage color scores in a Finnish population of polymorphic tawny owls (*Strix aluco*; Fig. 1B), we investigate the extent and possible role of intra-morph phenotypic variation for evolutionary change in a long-lived polymorphic bird of prey. In this species, plumage coloration is highly heritable and expressed in a gray and brown morph following a one-locus/two-allele Mendelian inheritance system with brown dominance[25,26]. From this discrete genetic basis, both morphs express considerable within-morph variation in plumage coloration (Fig. 1B), through which they might respond to environmental conditions. Specifically, a gray plumage is associated with higher cold tolerance and crypsis in snowy conditions, while brown plumage coloration is expected to be more adaptive to warmer winters with little snow[27]. Previous research indicated a

rapid shift in morph frequency from gray-dominated to roughly equal frequencies of both morphs[25], which is attributed to relaxed selection against the brown morph in milder snow-poor winters[25], and supported by recent work that identified selection on camouflage coloration and plumage characteristics as possible mechanisms[28–30]. However, thus far it has not been explicitly tested whether a changing climate with warmer winters leads to the expected unidirectional shift in plumage coloration across both morphs, i.e., toward more pheomelanic brownish phenotypes and away from gray (Fig. 1A).

Our comprehensive dataset, which also includes information on breeding behavior and performance, allows us to address these and other questions related to intra-morph phenotypic variation. Specifically, we test i) for the presence and dynamics of intra-morph variation in plumage coloration, ii) whether variation in plumage coloration affects breeding behavior across a range of winter conditions, and iii) whether developmental effects contribute to the observed divergence in plumage coloration. Overall, our study reveals substantial intra-morph variation in plumage coloration, whose dynamics are shaped by a complex interplay of morph-specific selection on life-history traits in breeding adults, and likely by differential

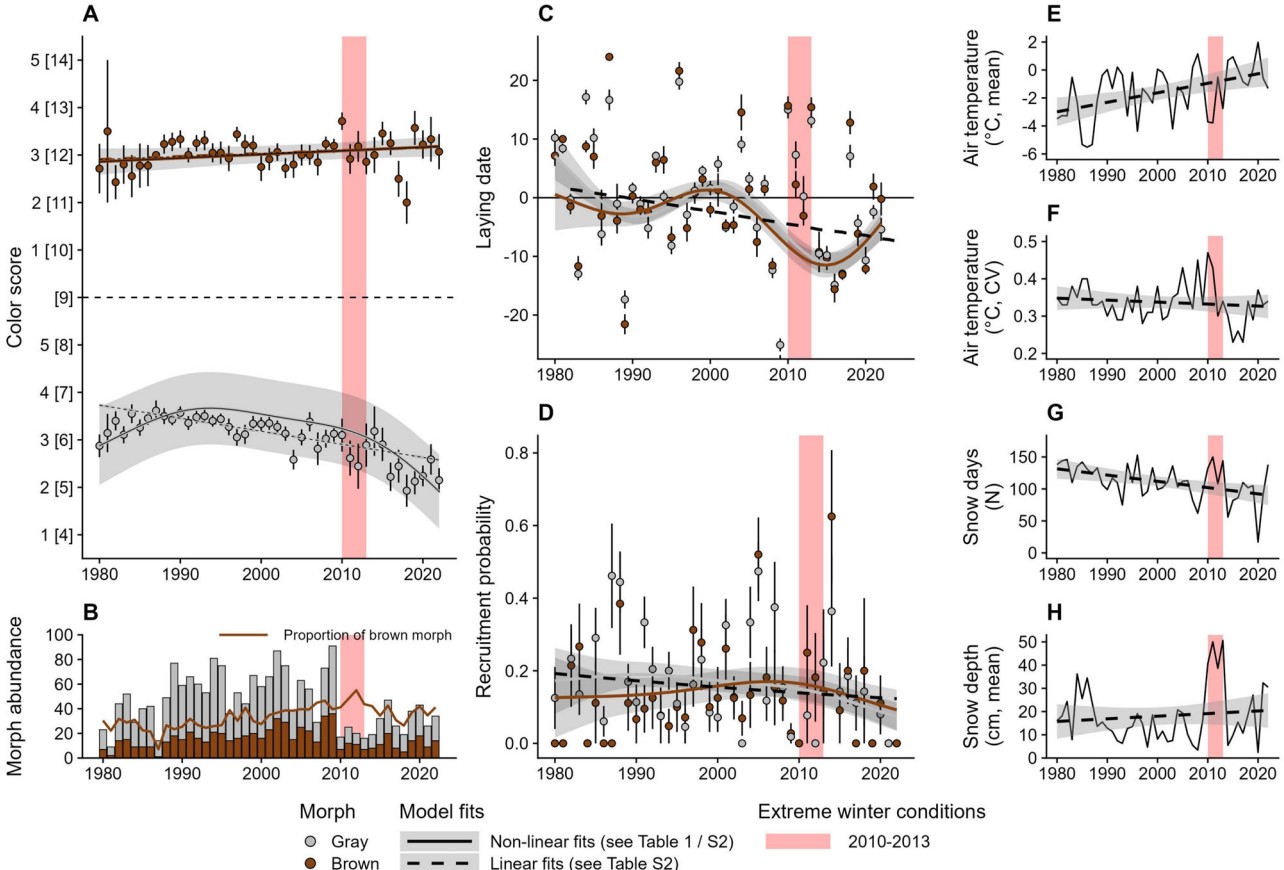

**Fig. 2 | Tawny owl study population between 1980 and 2022. A** Color scores of captured breeding adults are highly divergent over the years. Points denote the yearly average of morph-specific color scores using all breeding individuals per each year (mean ± SE, $N = 1972$). The solid lines are fits from a non-linear Generalized Additive Model (GAM 1; shading denotes ± CI of fit), the dashed lines are fits from a linear model (LM 1; Supplementary Table 2). The red annotation depicts a series of severe winters (panels E-H), which resulted in a dramatic reduction of captures (panel B). On the y-axis both relative and absolute (in square brackets) color scores are provided. **B** Abundances of all individuals per morph (bars) and morph frequencies/proportion of the brown morph over the years (brown line). **C** Average egg

laying date for each morph over the years, relative to median laying date March 31st (= 0). The solid lines are morph-specific fits ( ± CI) from GAM S2 and the dashed line is the fit from a linear model (LM 2; Supplementary Table 2 - main effect only). **D** Probability of recruitment of offspring born in the population over the years. Solid lines are morph-specific fits from GAM S3, and the dashed line is the fit from a linear model (LM 3; Supplementary Table 2 - main effect only). **E–H** Information on winter conditions expressed as the mean air temperature, coefficient of variation (CV) of air temperature, the sum of snow days, and mean snow depth. The solid black lines are fits of the respective terms in the path analysis (SEM 1).

sensitivities to genetic and environmental factors in juvenile owls that are highly morph-specific.

## Results

### Variation in intra-morph plumage coloration
Our study was based on long-term monitoring data collected between 1978 and 2022 from an area in southern Finland equipped with nest boxes. Using standardized ordinal color scores (controlled for repeatability and accounted for observer-based measurement error, see Supplementary Fig. 1 and Methods for details) from all usable breeding adult observations ($N = 1972$; Supplementary Table 1), we detected a significant divergence in plumage coloration between the two morphs over time (Fig. 2A), which was mostly driven by the gray morph. Specifically, a Generalized Additive Model (GAM 1; Table 1), showed no significant changes in the brown morph (only a marginally significant linear trend for higher scores), while color scores in the gray morph exhibited a hump-shaped pattern, with increasing values during the 1980s, slowly decreasing values between 1990 and 2010, and rapidly decreasing values since 2010, following the occurrence of extremely harsh winters (red annotations in Fig. 2). These morph-specific differences persisted after controlling for observer-specific effects on color scoring (Supplementary Fig. 2A), the contribution of immigrating individuals (i.e.,

the possible effect of gene flow, GAM S1a), and demographic cohorts (i.e., only first-time breeders vs the entire, cumulative population, GAM S1b; Supplementary Fig. 2B, C and Supplementary Table 2; see also LM 1 for comparison with GAM 1 in Supplementary Table 3).

### Heritability
We conducted a series of analyses to confirm that plumage coloration in tawny owls is highly heritable, as shown by a previous study ($h^2 = 0.79$[25],). First, we used a parent-offspring regression (POR) of 170 recruit- over midparent color scores, which showed a similarly high level of broad sense heritability ($H^2 = 0.83 ± 0.07$, fit ± SE; Fig. 3A and Supplementary Table 4). To corroborate this remarkably high heritability, we leveraged the pedigree information from our population (Supplementary Fig. 3) to implement an "animal model" following the specifications from Karell et al.[25]. This baseline model revealed a comparably high narrow sense heritability estimate (AM1: $h^2 = 0.89$; Supplementary Table 4), where the common environment effect—representing shared nest conditions—contributed negligibly to phenotypic variation (1.3%). This is in line with the POR result but likely inflated, as such models lack relevant environmental covariates and may attribute shared environmental variance to genetic effects. Indeed, after explicitly encoding the candidate

**Table 1 | Summary table for Generalized Additive Models (GAMs)**

| Model | Response variable | Component | Term | | Est. df | Ref. df | Chi.sq / F-value | p value |
|---|---|---|---|---|---|---|---|---|
| GAM 1 | Color score | Linear component | | Morph | | 1 | 0.352 | 0.5529 |
| | | Smooth terms | Year | Gray | 3.757 | 3.965 | 33.196 | <0.0001 |
| | | | | Brown | 1.001 | 1.002 | 3.387 | 0.0656 |
| | | | Observer | Gray | 1.958 | 2.000 | 83.979 | <0.0001 |
| | | | | Brown | 1.348 | 2.000 | 2.528 | 0.0254 |
| GAM 2 | Color score | Linear component | | Morph | | 1 | 0.152 | 0.697 |
| | | Smooth terms | Air temp. (Mean) | Gray | 1 | 1.001 | 5.986 | 0.0145 |
| | | | | Brown | 1 | 1.001 | 0.253 | 0.6156 |
| | | | Snow days (N) | Gray | 3.336 | 3.557 | 5.489 | 0.0028 |
| | | | | Brown | 1 | 1.001 | 0.747 | 0.3877 |
| | | | Air temp. x Snow days | Gray | 7.311 | 8.510 | 2.797 | 0.0037 |
| | | | | Brown | 1.000 | 1.001 | 0.703 | 0.4019 |
| GAM 3 | Laying date | Linear component | | Morph | | 1 | 3.382 | 0.0661 |
| | | Smooth terms | Air temp. (Mean) | Gray | 3.350 | 3.760 | 50.070 | <0.0001 |
| | | | | Brown | 2.234 | 2.749 | 35.946 | <0.0001 |
| | | | Color Score | Gray | 1.007 | 1.015 | 0.780 | 0.378 |
| | | | | Brown | 1.001 | 1.001 | 0.015 | 0.906 |
| | | | Air temp. x Color Score | Gray | 1.679 | 2.119 | 2.757 | 0.057 |
| | | | | Brown | 1.969 | 2.446 | 1.147 | 0.294 |
| GAM 4 | Recruitment success | Linear component | | Morph | | 1 | 1.456 | 0.2280 |
| | | Smooth terms | Laying date | Gray | 1.000 | 1.001 | 0.078 | 0.7806 |
| | | | | Brown | 1.018 | 1.035 | 2.223 | 0.1370 |
| | | | Color Score | Gray | 2.397 | 2.920 | 6.554 | 0.0688 |
| | | | | Brown | 2.013 | 2.507 | 1.905 | 0.4016 |
| | | | Laying date x Color Score | Gray | 1.001 | 1.001 | 1.745 | 0.1867 |
| | | | | Brown | 1.000 | 1.001 | 4.693 | 0.0303 |
| GAM 5 | Recruit color score | Linear component | | Morph | | 1 | 1.665 | 0.1989 |
| | | Smooth terms | Paternal color score | Gray | 2.495 | 3.012 | 3.882 | 0.0103 |
| | | | | Brown | 1.000 | 1.000 | 0.023 | 0.8792 |
| | | | Maternal color score | Gray | 1.301 | 1.504 | 6.425 | 0.0282 |
| | | | | Brown | 1.001 | 1.002 | 1.451 | 0.2297 |
| | | | Paternal color score x maternal color score | Gray | 3.646 | 11.000 | 0.889 | 0.0110 |
| | | | | Brown | 0.000 | 3.000 | 0.000 | 0.7261 |
| GAM 6 | Recruit color score | Linear component | | Morph | | 1 | 3.213 | 0.0746 |
| | | Smooth terms | Air temp. breeding (Mean) | Gray | 1.000 | 1.000 | 1.680 | 0.1964 |
| | | | | Brown | 1.000 | 1.000 | 4.431 | 0.0366 |
| | | | Air temp. breeding (CV) | Gray | 1.039 | 1.076 | 2.165 | 0.1300 |
| | | | | Brown | 1.947 | 2.394 | 1.700 | 0.1536 |
| | | | Air temp. Mean x Air temp. CV | Gray | 1.000 | 1.001 | 0.064 | 0.8018 |
| | | | | Brown | 1.000 | 1.000 | 5.292 | 0.0225 |

GAM 1 tests temporal trends in color scores (Fig. 2); GAMs 2–4 test interactive effects between adult color score and life-history traits (Fig. 4); GAMs 5–6 test genetic and environmental effects on recruit color score (Fig. 5). *Est. df* = estimated degrees of freedom; values near 1 suggest a linear effect, while higher values indicate potential non-linearity, but significance must be assessed from the *p* value column. *Ref. df* = reference degrees of freedom for the smooth term. *Chi.sq/F-value* = test statistic for smooth or linear terms.

genetic locus, its dominance effect, and the four key environmental variables (see Figs. 4, 5) in a second model (AM2; see "Methods" for details), the additive genetic contribution dropped by 66 percentage points ($h^2 = 0.23$), while the nest effect still had a minor impact (9.6%; Supplementary Table 4). These lower estimates likely reflect a more conservative and realistic assessment of heritability in this system, but due to the strong overlap between maternal identity, paternal identity, and breeding territory, as well as limited within-family replication (1.62 ± 0.09 recruits per dam, mean ± SE), it remains difficult to fully disentangle genetic from environmental contributions with our data.

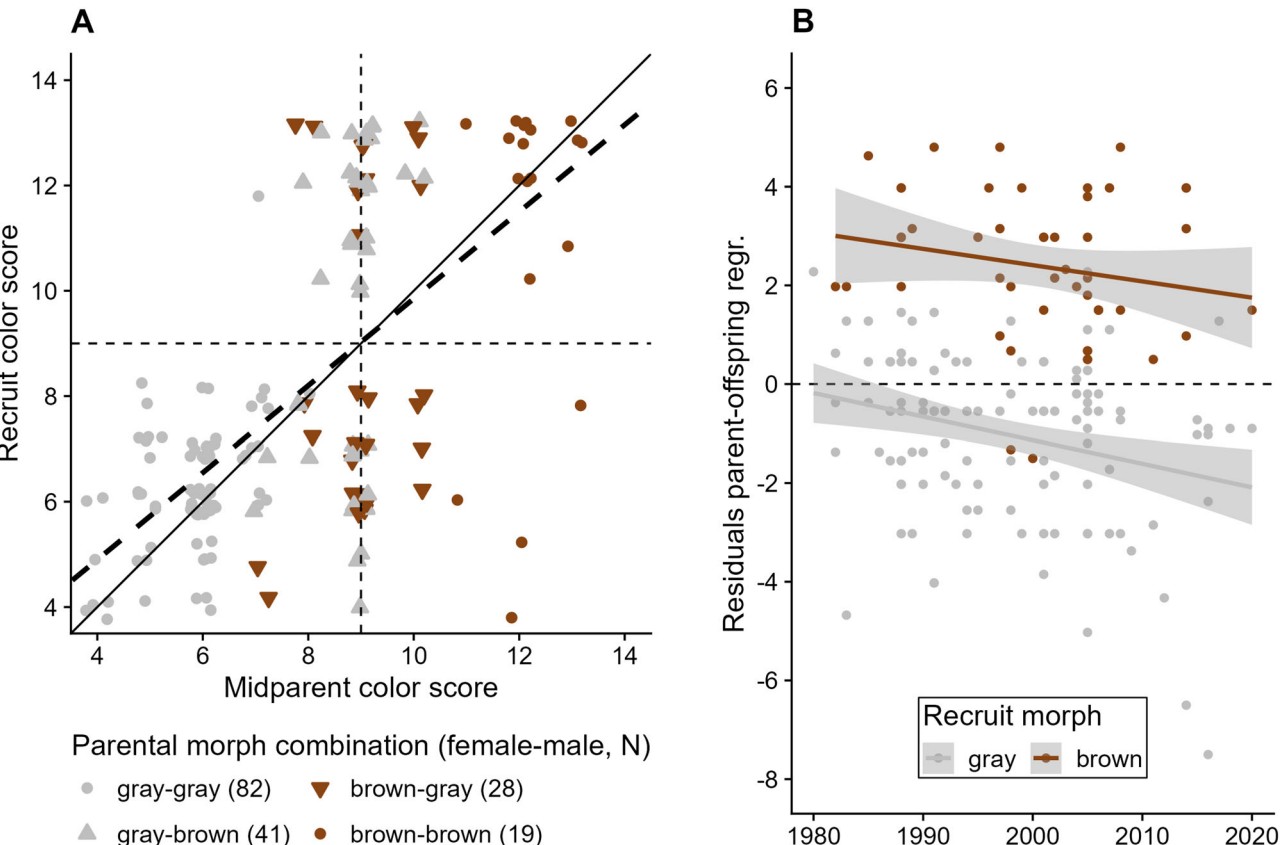

**Fig. 3 | Heritability of plumage coloration in tawny owls. A** Parent-offspring regression ($N = 170$), using the mid-parent values (i.e., mean pair color score) for absolute color scores. The grid denotes transition between morphs both for parents (*x*-axis) and recruit (*y*-axis), where color scores below 9 indicate gray morph, while higher color scores represent brown morph. Symbols depict different parental morph combinations. The thick dashed line denotes the regression line (coefficient = 0.825, $R^2 = 0.41$, $p < 0.0001$); the solid line is the 1-1 line, for comparison. **B** Residuals of the parent-offspring regression over time, grouped by recruit morph. Gray shading shows corresponding standard error.

We also assessed the stability of heritability over time by examining temporal trends in the residuals of the parent–offspring regression and the full animal model. Using GLS models with an autocorrelation structure, we found a consistent and significant decline in residuals across both morphs over the 40-year study period ([year]: POR: $t = -3.137$, $p = 0.002$; AM2: $t = -3.595$, $p = 0.0004$; [morph]: POR: $t = 4.542$, $p < 0.0001$; AM2: $t = 3.289$, $p = 0.0012$; Supplementary Table 5), indicating that the colour score similarity between parents and recruits has weakened over time (Fig. 3B). Importantly, the decline was parallel in both morphs, suggesting that this pattern reflects a general, population-wide process rather than morph-specific changes ([year*morph]: POR: $t = 1.157$, $p = 0.249$; AM2: $t = 1.548$, $p = 0.1235$; Supplementary Table 5). Such a reduction in parent–offspring resemblance could result from increasing environmental variance, changes in selection regimes, or shifts in the distribution of genotypes over time, any of which would reduce the proportion of phenotypic variance attributable to additive genetic effects.

### Effects of plumage coloration and breeding phenology on adult fitness

To assess whether and how morph-specific coloration divergence correlates with breeding performance, we examined color scores in relation to laying date (March 31 = 0; $N = 1866$ after removing missing values and outliers; Supplementary Table 1) and recruitment success (i.e., the proportion of new breeders originated from known individuals within the monitored population) as a proxy for fitness. Across both morphs, the owls in the study population have responded to increasingly warm winters with little snow by advancing their average laying date by 10 days over the observed period (Fig. 2C; both morphs: $p < 0.0001$, GAM S2; LM 2; trend: gray = −0.202, t-

ratio = −6.043, $p < 0.0001$; brown = -0.222, t-ratio = -4.687, $p < 0.0001$, Supplementary Tables 2, 3 and 6). At the same time, the probability of successful recruitment (i.e., new first-breeding adults in the population) has decreased from just under 20% between 1980 and 2010 to almost 10% between 2010 and 2020 across both morphs (Fig. 2D). However, there is a differential between the barely negative trend in the brown morph and the slightly more negative trend in the gray morph shifting the balance towards more new brown breeders over time (Fig. 2D; gray: $p = 0.0163$, brown: $p = 0.2632$, GAM S3; LM 3; trend: gray = -0.021, t-ratio = -2.929, $p = 0.0034$; brown = -0.004, t-ratio = -0.36, $p = 0.7191$, Supplementary Tables 2, 3 and 6). To investigate possible relationships between broad scale changes in winter conditions (i.e., half a year timeframe—see Supplementary Fig. 4), advancing laying date, and differential recruitment success in relation to color score, we conducted a path analysis using Structural Equation Modelling (SEM 1). Using all observations with known laying date ($N = 1866$), we found weak and marginally significant effects (i.e., path coefficients; Fig. 4A) of snow on color score (more "grayness" with more snow; [snow depth]: $z = -3.079$, $p = 0.0021$; [snow days]: $z = 1.880$, $p = 0.0601$) but no direct effect of color score on either laying date ($z = 1.299$, $p = 0.1941$) or recruitment success ($z = -0.052$, $p = 0.9584$; Fig. 4A). The path analysis also revealed that temperature had the highest impact on laying date (~2 days earlier for every degree of warming; $z = -6.207$, $p < 0.0001$), while the number of snow days ($z = 2.172$, $p = 0.0298$) and snow depth ($z = 3.534$, $p = 0.0004$) had only small effects. However, laying date was not associated with variation in recruitment success ($z = -1.056$, $p = 0.2911$), which slightly decreased in snow rich winters ([snow depth]: $z = -2.61$, $p = 0.009$; Fig. 4A; Supplementary Table 7 for full model output and estimates).

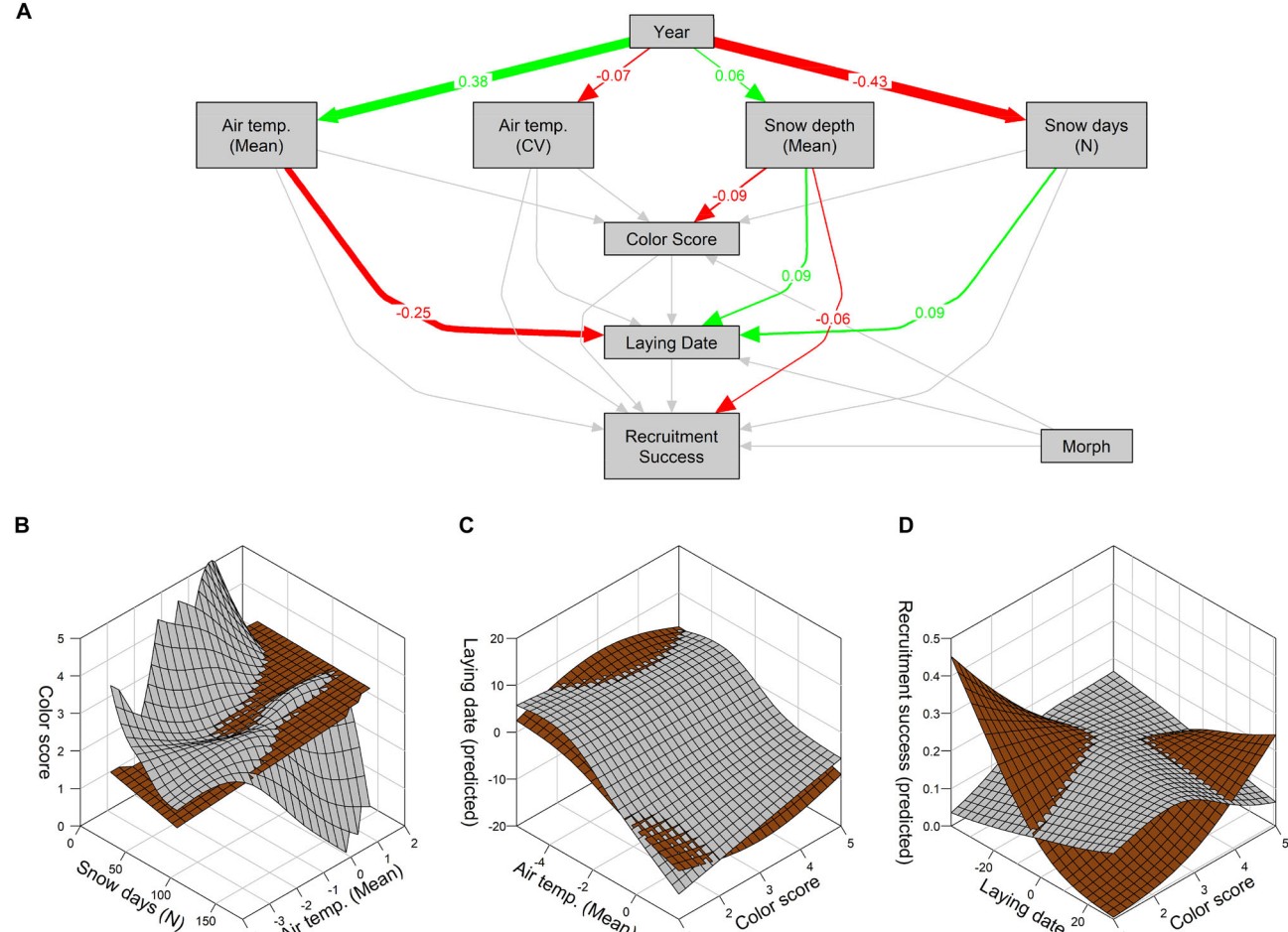

**Fig. 4 | Selection on plumage coloration in breeding adults. A** Path analysis based on Structural Equation Modelling (SEM 1) that includes all observations of breeding adults with known laying date ($N = 1866$, 80.5% of the entire dataset), indicating a strong effects of a changing environment on laying date. Colored arrows indicate significant positive (green) or negative (red) effects; gray arrows are not significant effects. The number and width of an arrow show the effect size. Model summary is in Supplementary Table 4. **B** Test for interactive effects of morph, snow days and mean winter temperature on the color score of breeding adults using a Generalized Additive Model (GAM 2). The model shows that gray individuals were more susceptible to changes in winter conditions. **C** Test for interactive effects of morph, mean winter temperature and color score on laying date (GAM 3) revealed no differences between morphs in breeding behavior according to color score. **D** Test for interactive effects of morph, color score and laying date on recruitment probability (GAM 4). In the brown morph, a close match between breeding adult color scores and laying date increased the likelihood of their offspring recruiting back into the population. In the gray morph, there was a main effect of color score, but no interaction with laying date.

Since SEMs allow only for tests of linear relationships without interactions, we used a set of three GAMs (GAM 2-4; Table 1 and Fig. 4B, C) to test for possible interactive effects between color phenotype, breeding performance, and fitness. In GAM 2, we used an interactive term with temperature and snow days to assess whether, at the population level and across the entire observed period, color scores of breeding adults were associated with specific environmental conditions. Indeed, the model revealed a non-linear response in the gray morph to variation in winter conditions, but not in the brown morph (Fig. 4B). Cumulatively, i.e., over all observed years, the lowest color scores (i.e., more gray dominated plumage) were associated with mild winters but high precipitation, i.e., many snow days, the highest color scores (i.e., increasingly brown plumage) were associated with mild winters with little precipitation, and intermediate color scores forming a plateau between the two environmental maxima. In GAM 3, we used an interactive term with temperature and color score to test whether an individual's color phenotype would affect the decision when to breed in relation to environmental conditions. We found that, for both morphs, laying date was mostly determined by winter temperature and not by plumage coloration, suggesting that the decision for when to breed is formed independently of the color phenotype (Fig. 4C). In GAM 4, we tested whether the

interaction between color score and laying date could predict recruitment success, testing the central hypothesis that light plumage coloration has fitness benefits at earlier laying dates, while darker plumage coloration is expected to be more beneficial when breeding later in the season[27]. We found support for this hypothesis in the brown morph, where recruitment success was higher when the color score matched the laying date, i.e., when the individual's coloration was light at earlier laying dates and more intense at later laying dates (Fig. 4D). In contrast, in the gray morph, there was no such interaction and the probability of producing new breeders was only marginally determined by color score, independent of laying date.

## Genetic vs environmental effects on plumage coloration in recruits

We investigated the interplay of genetic factors (parental morphs and color scores) and environmental effects (in the period between incubation and fledging) on the expression of plumage coloration in the offspring using a second path analysis (SEM 2). This analysis used only the pedigree ($N = 170$) and the same four environmental parameters as in SEM 1, but measured only during the breeding period (i.e., 56 days following laying date, which is the average breeding period for this population - see Methods for details).

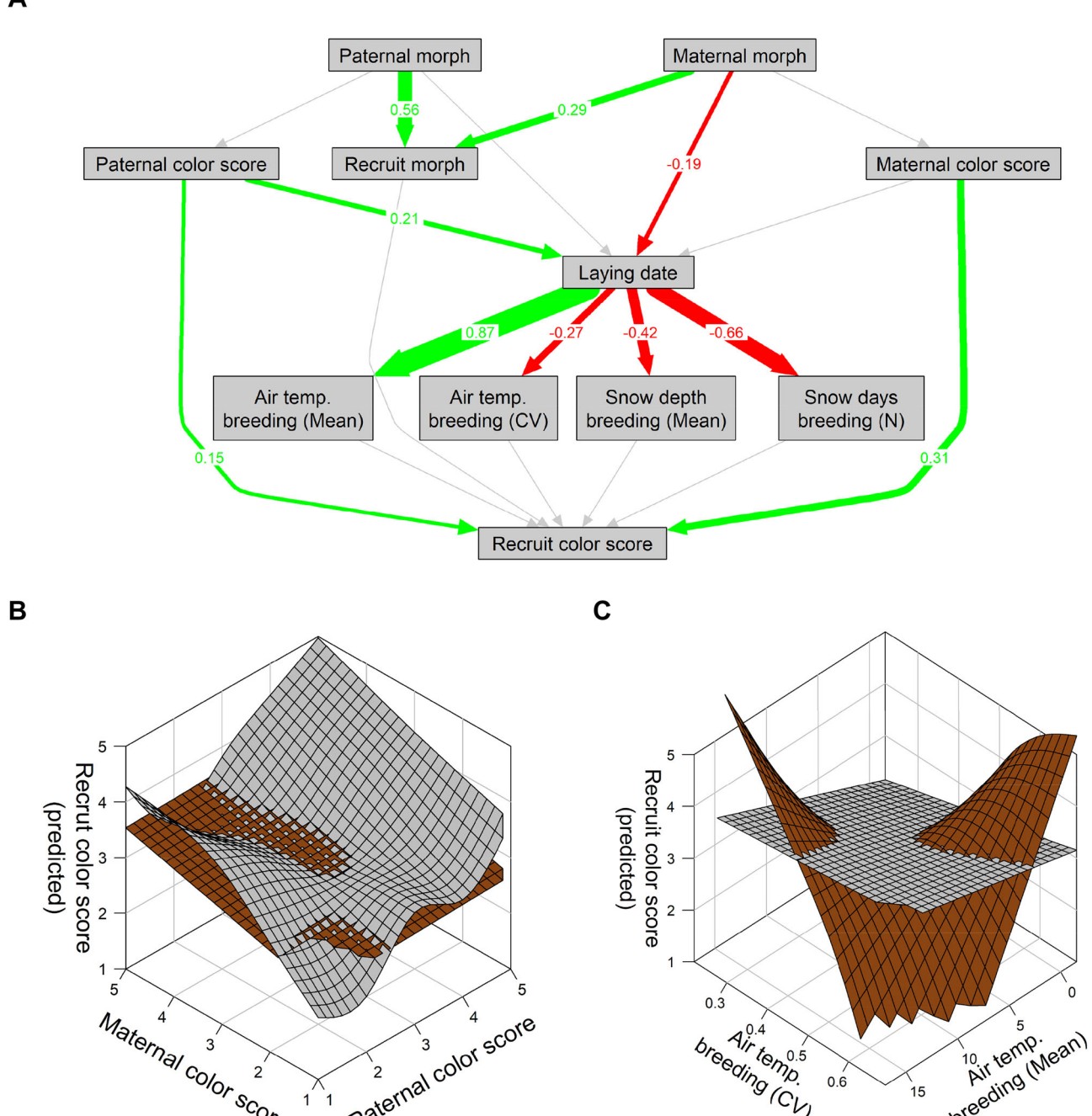

**Fig. 5 | Genetic and development effects on recruit color score. A** Path analysis testing for genetic (i.e., paternal and maternal morph and color score) and environmental factors (i.e., laying date, snow depth and temperature before fledging) on recruit color score ($N = 170$), using a Structural Equation Model (SEM 2). The path network should be interpreted in the same fashion as the one shown in Fig. 4. Overall, environmental conditions during the breeding period were strongly associated with the timing of the laying date itself, hence the high effect sizes. **B** Model fits from GAM 5 that uses paternal and maternal color score as predictors of recruit color score. Parental color scores only affect recruit color scores in the gray, but not in the brown morph. **C** Recruit color score model fits from GAM 6 ($N = 206$) that uses mean and coefficient of variation of temperature between laying date and fledging as predictors. The model suggests that brown morph recruits are more strongly affected by environmental variation than the gray morph.

The second path analysis further corroborates the finding that plumage coloration in tawny owls is a highly heritable trait (Figs. 3A and 5A and Supplementary Tables 4 and 8). Additionally, the analysis indicates a higher contribution to recruit color score from the maternal ($z = 4.466$, $p < 0.0001$) than from the paternal ($z = 2.164$, $p = 0.030$) side, hinting at the possibility of a sex-linked genetic mechanism or maternal effects on coloration. The path analysis also revealed moderate effects of paternal color score ($z = 2.991$, $p = 0.0028$) and maternal morph ($z = -2.71$, $p = 0.0067$) on laying date

(Fig. 5A and Supplementary Table 8 for full model output), which, however, might be due to the lower sample size compared to all breeding adults in SEM 1.

We used two GAMs (GAM 5–6; Table 1 and Fig. 5A–C) to test for interactive effects of either parental phenotype or environmental effects during breeding on recruit color score. Specifically, GAM 5 included an interactive term for both parental color scores and showed an interactive effect of parental coloration on recruit color scores in gray but not brown

recruits. This suggests that parental determination of coloration is stronger in the gray than in the brown morph, thus indicating a higher contribution of environmental effects to coloration in the brown morph (Fig. 5B). Further supporting this, GAM 6, which tested for effects of environmental variation during early development on color score, revealed a significant interactive effect of mean temperature and temperature variation (i.e., temperature CV) during the breeding period, i.e., between laying date and fledging in the brown morph, but not in the gray morph (Fig. 5C; breeding period = 56 days [average for the study population]).

## Discussion

Our analysis of 43 years of field observations ($N = 1972$) indicate substantial within-morph variation in plumage coloration of tawny owls, which was unexpectedly highly divergent between the two morphs (Fig. 2). On the one hand, following the general expectation of increased pigmentation at higher temperatures (i.e., Gloger's rule)[31–33], we found increasingly intense pheo-melanic (i.e., more fulvous) coloration in the brown morph. On the other hand, and deviating from our expectation, the gray morph followed an opposite trend towards lower color scores (i.e., more gray-dominated plumage). Our findings suggest that these contrasting changes within the two morphs in a highly heritable trait are likely driven by a combination of morph-specific trade-offs between life-history strategies and plumage phenotypes[26], differential hereditary dynamics, and, putatively, reduced adaptive capacity and plasticity in the gray morph compared to the brown one.

Rapid phenotypic divergence within populations typically occurs when intermediate phenotypes are selected against in the prevailing environment[34,35]. In tawny owls, warmer winters appear to favor the melanistic brown morph over the gray[25], but it remains unclear what life-history traits could contribute to an increase in frequency of the brown morph under warming winter conditions. Brown parents produce fledglings that are in better condition than those of gray parents[26]. We therefore explored the variation in recruitment success in relation to plumage coloration, and found that while breeding timing was largely decoupled from adult coloration of both morphs (Fig. 4C), the probability of successful recruitment was not (Fig. 4D): in the brown morph, recruitment into the breeding population was predicted by a close match between laying date and plumage coloration, which, following warmer winters, would select for increasingly pigmented recruits. In the gray morph, however, there was no significant interactive effect of color score and laying date, and the main effect was only marginally significant, suggesting a weak trend for intermediate color scores to enhance recruitment success across laying dates. Accordingly, intermediate gray adults might have a higher chance of coping with variable environments than either end of the phenotypic spectrum. Such capacity might be advantageous given the gray morph's longer lifespan and higher lifetime reproductive output[36], which increase the exposure to environmental variability. Additionally, the gray morph shows a more flexible breeding effort than the brown morph, and typically refrains from breeding under unfavorable climate[37].

The genetic composition of a population can also be altered by stochastic events that reduce the phenotypic variation natural selection can act upon[38]. In 2009, the size of the breeding population dropped by about 80% (Fig. 2) because of very harsh winter conditions characterized by low temperatures, many snow days, and high snow depth that continued until 2013, and simultaneous crash in vole populations[39]. After this event, the population did not recover, suggesting increased mortality and a sharp reduction in standing genetic variation. Such bottlenecks may occur without changing morph frequencies over time directly[40], which was also not the case in our population, as the relative proportion of brown individuals in the population kept increasing in those years (Fig. 2B), but indirectly by reducing the available heritable phenotypic variation, and thus the potential to respond to future selection pressures[41]. Plausibly, the series of severe winters has selected for lower plumage pigmentation in gray individuals, which would explain the sharp drop in average color scores after 2013. Following these events, the gray morph may have lacked sufficient genetic variation to

respond to selection for higher production of pheomelanin under the long-term trend of warmer winters with less snow[25]. This hypothesis is supported by a recent genomic study that isolated the genomic basis of cold tolerance in the gray morph and also identified temporal shifts in the allelic frequencies of the candidate loci's genotype towards an excess of homozygotes, i.e., individuals of the gray morph, hinting at the possibility of overall stronger selection among gray owls[27]. Different dispersal behavior that can replenish genetic diversity of populations via gene flow[42] may also play a role here: previous research showed that the two morphs on average disperse similar distances, but the gray morph travelled further in colder winters, while the brown dispersed further in warmer winters[43]. Under increasingly milder conditions, therefore, we would expect local phenotypic variation to increase at a higher pace in the brown morph than in the gray.

Our findings suggest morph-specific differences in the heritable components of plumage coloration as a potential driver of phenotypic divergence. Regardless of the morph, our analyses confirmed an overall high heritability of plumage coloration in tawny owls[25,44]. However, in the gray morph, we additionally found that parental contribution was significantly stronger, while plumage coloration of brown recruits was to a significant extent affected by environmental conditions experienced during early-life period (i.e., pre-fledging). Together, these results point to the possibility of morph-specific genetic and non-genetic transgenerational effects, such as carry-over effects due to environmental conditions[45], which may shape quantitative variation in plumage color. Sources of non-genetic variation may include environmental effects in the parents, such as differential parental efforts, particularly in the female, as suggested by the higher maternal effect, where food allocation and reproductive effort may vary according to performance of parental phenotypes[46,47]. Unfortunately, the shallow structure of our pedigree data prevented us from robustly estimating the variance component explained by maternal effects, which may have led to inflated heritability estimates. In the brown morph, possible non-genetic effects may be induced by pre- and postnatal environmental conditions, as we observed for temperature during breeding in the brown recruits, which hints at developmental plasticity. Among the four environmental predictors in the full animal model, only temperature CV showed a positive, although marginal effect on plumage coloration ($p = 0.0605$; Supplementary Table 4). While not statistically significant at conventional thresholds, this trend suggests that recruits exposed to more variable thermal conditions during development might exhibit environmentally-induced variation in plumage coloration consistent with phenotypic plasticity. Phenotypic plasticity in pigmentation, the ability of a genotype to produce different pigmentation phenotypes in response to the environment, is often observed in invertebrates[48–50], but observations from non-model vertebrates and birds are less common[13,51]. Specifically in polymorphic species, morph-specific plasticity is known for butterflies[52] and lizards[14,53]. In tawny owls, plasticity in plumage pigmentation might allow the brown morph to trace optimal phenotypes under a changing environment more quickly while buffering extremes[14,53], which might be reflected in both the increasing frequency of the brown morph[25], but also the relative steadiness of the increase in color scores through the years (Fig. 2A). We urge caution when interpretating these results pertaining heritability, and recommend further experimental research to elucidate the role of temperature variation in affecting pigment deposition in early-life stages[48].

At its core, our study provides evidence for differential, morph-specific production of pigmentation in polymorphic populations of owls. This finding adds complexity to the prevailing idea that CPs are expressed through distinct genetic pathways, such as pigment-type switching systems, which govern the type of melanin to be produced among morphs[14,22,54]. While our finding does not change the value of CPs as markers of (discrete) genetic variation[2], it does open up more questions, highlighting the possibility of environmental effects on and physiological limits to phenotypic change. Importantly, the presence of continuous variation may affect the evolutionary response to a changing environment: individuals who do not produce or produce only partly the pigment putatively conferring benefits under changing environmental conditions might fail in tracing new

phenotypic optima (Fig. 1). This suggests that in addition to an overarching Mendelian genetic architecture that determines morph, additional quantitative trait loci and environmental factors may control pigment deposition and contribute to individual variation in continuous coloration. Current knowledge of the genomic underpinnings of CP in tawny owls highlights important structural variation in the melanin concentrating hormone receptor gene (MCHR), which codes key physiological responses involved in adaptation to cold[27]. Thereby, evidence does point towards gradual variation in physiological profiles, especially in the gray morph. Moreover, the diverse molecular origin of pheomelanin coloration suggests that its exact genetic architecture may be more complex than phenotypic data alone seems to suggest[25]. Overall, the role of genetic architectures interacting with the environment during development remains unclear and requires further research, particularly in the context of CPs and intra-morph phenotypic variation.

Despite the importance of genetically-based CPs for biological diversity[9,10], explicit consideration of intra-morph color variation remains limited in the study of evolutionary processes[8,9]. Based on our findings, we argue that quantifying intra-morph phenotypic variation can increase our understanding of how polymorphisms are maintained[55,56], but also how populations persist or change in space and time[21,57]. On the one hand, the substantial intra-morph variation we detected implies that polymorphic species can host larger phenotypic diversity than previously thought, which may increase the evolutionary potential of populations to respond to environmental change[41,58,59]. On the other hand, since each morph can become limited in the amount of heritable phenotypic variation, gene-switch-based polymorphisms may also increase the likelihood of producing maladapted forms with reduced adaptive capacity, which might decrease the stability of the polymorphism within the population. Specifically, while the brown morph continues to produce darker phenotypes that have higher fitness in a warming environment, the gray morph appears to be "locked in" at low plumage pigmentation following a series of harsh winters. This might, for instance, reduce the competitive abilities of the gray morph under a warming environment. Gene flow into the population may be a potential mechanism in the gray morph to produce more pigmented individuals, as we found that in gray immigrants color scores tended to decrease over time but not as strong as in the resident breeding pool (Supplementary Fig. 2B). Overall, gauging how these contrasting effects shape the persistence of polymorphic populations remains challenging, and more long-term studies are needed to better understand how continuous variation within CPs contributes to evolutionary processes.

## Conclusion

In summary, our study shows that fitness optima within the two morphs might be contingent on changes in continuous coloration associated with climatic conditions. Importantly, we found first support for a stronger environmental effect on the phenotype of the brown morph, which has likely promoted phenotypic divergence in the study population under a warming climate. In contrast, the gray morph might be more constrained by its genetic architecture and be less efficient in tracking environmental changes. These temporal changes in quantitative traits have the potential to influence evolutionary dynamics by either preventing morphs from outcompeting each other or by changing morph frequencies to the point where one morph is lost. Our findings thus raise important questions on how CPs might be maintained, among other mechanisms, through changes in the distribution of continuous phenotypes within morphs.

## Methods
### Study area and field protocol
We used data from a population of tawny owls that was collected between 1978 and 2022. The population is monitored using nest boxes and extends over an area of about 500 km$^2$, comprising two almost overlapping and equally sized areas in western Uusimaa, Southern Finland (60°15' N, 24° 15' E). The area is equipped with ca. 300 nest boxes (about 150 each). Part of the area was monitored from 1978 onward, and the other one was monitored

from 1987 onward (for description of the study area, see refs. 26,43). In Finland, tawny owls live in mixed and boreal forests where they promptly breed in nest boxes from late February to late April (median laying date 31 March). In April, all nest boxes were checked to detect breeding attempts and record information on clutch size as well as hatching date. Upon nest inspection, laying date was estimated retrospectively by counting from hatching or by estimating the age of the chicks from their wing length if they had already hatched. During the nestling period, both parents were trapped at the nest box, aged, measured, and ringed (if not ringed before) to allow individual identification[39]. Males and females have similar capture probability[36,39]. We have complied with all relevant ethical regulations for animal use, and all birds were captured, handled, and ringed with an appropriate ringing license.

### Plumage color scoring
The color scores used for studying tawny owls' color variation were assigned in the field upon capture, following a simple, standardized, and repeatable protocol that remained unchanged over the 43 years considered in this study[36,60]. The scoring protocol focuses on the degree of redness in the plumage, using a continuous semi-ordinal scale ranging from 4 to 14, where a higher score indicates more intense pheomelanic brown pigmentation in feathers, whereas, low score indicates gray dominated plumage without any pheomelanic component. The color score is calculated by summing up partial scores made for three different body parts: facial disc (1–3 points), breast (1–2 points), back (1–4 points), and general appearance (1–5 points)[36]. After scoring, the morphs are assigned based on the overall color scores (4–9 = gray morph, 10–14 = brown morph)[25,36]. Color scoring effort was constant in time, with only three different observers assigning color scores throughout two sub-populations (Supplementary Fig. 2). The first sub-population was monitored from 1978-2014 by a single person (observer "a") who, after thorough teaching and knowledge transfer, passed on all observation-duties (to observer "c") in 2013. The second sub-population was monitored by a single person from 1987 onward (observer "b"). When owlets fledge it is possible to determine the morph, but not a definitive plumage color score that the individual will have throughout its lifetime because the plumage is not fully developed yet[25]. Color scores were therefore only assigned to adults upon capture during breeding season, i.e., when individuals were recruited back into the population as breeders. Since tawny owls are long-lived and territorial birds that can occupy the same nest box for many years, a large portion (72%) of all individuals were captured and scored multiple times by the same observer (Supplementary Fig. 1A). In these cases, color score assignment was performed de novo, i.e., observers assigned a score without knowing previous color scores to avoid measurement bias. Using this protocol, color scores were highly repeatable over time[36] (see also Supplementary Fig. 1), which suggests that intra-individual variation is small and the between-year assessment of the observers reliable.

### Data pre-processing
Prior to all statistical analyses, we filtered unusable observations from the original dataset (Supplementary Table 1). Specifically, we removed individuals which showed excessive variation in color scores (standard deviation > 1.5; Supplementary Fig. 1B; $N = 19$). Since there are no reports of ontogenetic changes in plumage coloration in tawny owls, we considered within-individual variation as observer-based measurement error. The variation stemming from this error was small (Supplementary Fig. 1C), and because we were interested in long-term population dynamics, and not potential within-individual dynamics (e.g., due to resource availability[61]), we decided to remove it entirely by replacing all measured scores with their median. To confirm that raw- and median scores do not provide fundamentally different results, we ran the regression testing for long-term population dynamics with either (see below), but then used median color score. Hence, all presented analyses using color score refer to the per individual median color score. Moreover, when a color score record of an individual was incomplete, i.e., when a repeatedly scored individual was

lacking scores for one or more years, we used the median value of all collected scores to replace all values, including the missing ones. If only a single value was present in an individual's record despite having been captured several times, we used this to replace all missing values. Finally, due to the established color scoring protocol allowing for different ranges of possible color scores between the morphs (gray = 6 values, brown = 5 values), we removed all observations with a median color score of 9 (intermediate individuals, $N = 67$) from the dataset so that both morphs had the same number of possible color scores. This procedure also allowed further removal of individuals whose morph assignment might have changed over time. To ensure equally distributed data and homogeneity of variances for statistical analyses across both morphs, we transformed the color scores from an absolute scale to a relative scale ranging from 1 to 5 (Fig. 2A), to which we just refer as "color score" in the main text and in all figures.

## Statistical analyses

We conducted all statistical analyses in the statistical programming language R (v4.2.3[62],). All models used a Gaussian family, except the models with recruitment success (values of either 0 or 1) as the dependent variable, which used the binomial family (models LM3, GAM S3, and GAM 4—see below). All GAMs used thin plate regression splines for the smooth terms, 5 knots for the basis function, and restricted maximum likelihood (REML) for the smoothing parameter estimation.

**Temporal variation in intra-morph plumage coloration.** To test for differences in intra-morph variation in plumage coloration we used Generalized Additive Models (GAM; *mgcv* package v1.8-42[63],), which are highly suitable to model both linear and non-linear patterns in time series. We implemented a single additive model (GAM 1; Table 1) to test for the presence of morph specific variation, and to visualize the dynamics of color score over time. The GAM included morph as linear term ("parametric coefficient") and year by morph as non-linear term ("smooth term"), as well as the observer (see "Plumage scoring" section) as random intercept for the smooths to control for morph assignments made by different observers (Supplementary Fig. 2A). To confirm the validity and robustness of using the median-corrected color scores (see above), we fitted GAM 1 with the raw, uncorrected scores, and found equivalent results (not included, but can be reproduced from the enclosed data). Furthermore, we used two additional GAMs to test for the differences in color score dynamics between residents and dispersing/ migrating individuals (GAM S1a; Supplementary Table 2), and for differences between the entire population and the new cohort of each year (GAM S1b; Supplementary Table 2), with otherwise unchanged formulas and parameters. Additionally, we implemented a linear model to compare the non-linear estimates to linear ones (LM 1; Supplementary Table 3). Finally, we implemented a set of two complementary analyses to visually compare the temporal dynamics of color score change in the two morphs to average laying date (Fig. 2C; GAM S2; LM2; Supplementary Tables 2, 3 and 6) and breeding success (Fig. 2D; GAM S3; LM3; Supplementary Tables 2, 3 and 6).

**Heritability.** We conducted a parent-offspring regression (POR; Supplementary Table 4) using a linear model with mid-parent color scores (i.e., mean between parent color scores) to estimate heritability of plumage coloration (*stats* package v4.3.3[62],). Moreover, to confirm the robustness of this analysis, we implemented two 'animal models' (*sommer* package v4.4.1[64],): one simple model following the specifications from Karell et al.[25] (i.e., only including the intercept; AM1), and one more complex, where we encoded genotypes to assess the contribution of additivity and dominance at the genetic locus and the same key environmental predictors included in main analyses (mean temperature, temperature CV, snow cover, and snow depth; AM2). Specifically for additive and dominance effects, we coded parental morph combinations assuming their expected contribution to offspring genotype. On the one

hand, for additivity, the coding assumed an additive scale where the offspring's genotype effect on their coloration is the sum of "brown" alleles. Accordingly, the gray morph was coded as 0 and the brown as 1, so recruits from gray-gray pairs were assumed to have no brown alleles (=0), heterozygotes (i.e., recruits from mixed pairs) have 1, and recruits from brown-brown parents have 2. On the other hand, since dominance typically flags the heterozygous condition, we coded heterozygotes as 1 (dominant effect), and homozygotes (i.e., recruits from parents of the same morph) as 0. In both models, we included the nest ID as a random factor to account for possible common environment effects (e.g.,[65]). We were not able to reliably estimate maternal contributions to variance components, as within-dam replication and recruitment are low in the study population[26]. Because tawny owls are a territorial, site-tenacious species, and pairs typically use the same nest along their lifetime[66], it is challenging to disentangle maternal and other environmental effects. However, extra-pair paternity is uncommon in tawny owls[67], and we were confident in inferring relatedness on the basis of the social pedigree. In addition, tawny owls showed random mating with respect to coloration[36], thereby, we do not expect any bias in our estimation of the genetic component due to assortative mating[68]. To check whether heritability was temporally stable over the observed period, we tested the residuals of the POR and AM2 as a proxy of transgenerational changes in morph specific coloration and putative non-genetic effects using generalized least squares regression with an autocorrelation term (Supplementary Table 5; *nlme* package v3.1-164[69]).

**Relationship between plumage coloration, breeding performance, and success in adults.** We used Structural Equation Modelling (SEM; *lavaan* package v0.6-15[70]) to investigate the relationships between plumage coloration, breeding traits, and environmental variables in the adult breeding population within a single path diagram ($N = 1866$; SEM 1; Supplementary Table 7). Prior to constructing the path diagram, we used a sensitivity analysis (Supplementary Fig. 4) to determine that a 180-day window, i.e., the entire winter period, was best suited to model the effect of environmental variables on plumage coloration. Following this, the path diagram included time (year) and environmental variables between Nov. 1st and April 30th (air temperature [mean, hereafter "temperature"], air temperature variation [coefficient of variation, hereafter "temperature CV"] as a proxy of extreme events, snow depth [mean], snow days [N]), as well as morph, on color score, laying date, and recruitment success in the breeding population. The information on environmental variables was collected by the Finnish Meteorological Institute (FMI) at the weather station near Helsinki-Vantaa airport, situated *ca.* 50 km east of the study area (hourly measurements, aggregated by us to a daily mean). Laying date in our dataset sets March 31st as the median laying date we used as the reference (0) across years. Deviations from it take positive or negative values, accordingly (e.g., March 20th = −11, April 3rd = 3). Recruitment success is scored as 1 if any number of fledglings of a breeding pair of adults is recruited back into the population (i.e., appears again in the database), and 0 if not. Morph is coded as a binary variable (0 = gray, 1 = brown), and a positive value would indicate that brown has a stronger positive or negative effect than the gray on the variable it points at. For more comprehensive analysis of three key metrics in adult owls, and to test for interactive effects among variables (not possible in SEM-based path analysis) we implemented three additive models, all of which included a morph-specific interaction term: GAM 2 tested for interactive effects of air temperature and snow days on breeding adult color score, GAM 3 tested for interactive effects of air temperature and snow days on laying date, and GAM 4 tested for interactive effects of laying date and color score on recruitment success (see Table 1 for details).

**Genetic vs environmental effects on plumage coloration in recruits.** We used a second path analysis to investigate the effects of parental morph and color score, as well environmental dynamics during the

breeding period, on recruit color score (SEM 2; Supplementary Table 8). For this analysis we only used the records of recruits with both parental morphs known, color score, and laying date ($N = 170$; Supplementary Table 1). As a proxy of environmental changes during the rearing phase, we included mean temperature, temperature CV, snow depth, and snow days in a 56-day window after the laying date. This was the average time in our population between laying eggs and the timing of the oldest chick in a brood turning 25 days, shortly prior to fledging. For more comprehensive analysis of parental color scores, as well as environmental conditions on recruit color score, we implemented two additive models, both of which included a recruit-morph-specific interaction term: GAM 5 ($N = 170$) used an interactive term with maternal and paternal color scores to test differential heritability between morphs and paternal or maternal side. GAM 6 ($N = 206$) used an interactive term of mean temperature and temperature variation (i.e., temperature CV) to test for morph specific susceptibility for environmental effects on color score development.

### Reporting summary

Further information on research design is available in the Nature Portfolio Reporting Summary linked to this article.

### Data availability

We have archived all data necessary to reproduce the results and figures in an online data repository: https://doi.org/10.5281/zenodo.15392703[71].

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

## Acknowledgements

This paper is dedicated to the memory of Kari Ahola, who passed away during the writing of the manuscript. Without him, this and many other projects would not have been possible. We greatly thank Teuvo Karstinen and the other members of KBP for the many hours spent conducting fieldwork and data sharing. We also thank Glaucia Del-Rio, Arthur Porto, and Edward Iwimey-Cook for commenting on a draft of the manuscript and advice on animal models. The work was supported by the Swedish Cultural Foundation (grant numbers 168034 and 188919 to P.K.). A.P. was supported by a Margarita Salas fellowship from the University of Seville (MSALAS-2022-22312). M.D.L. was supported by a Marie Sklodowska Curie Individual fellowship awarded by the European Commission (PhenoDim; Grant No. 898932). This is paper number 25 from Kimpari Bird Projects (KBP).

## Author contributions

A.P., M.D.L., and P.K. conceived the study. E.A. and P.K. collected the data in the field. A.P. and M.D.L. analyzed the data. A.P. and M.D.L. wrote the first draft with inputs from P.K. All authors (A.P., M.D.L., E.A., and P.K.) revised and commented on the manuscript and approved the final version.

## Funding

## Competing interests

The authors declare no competing interests.
