## [Transparent Peer Review file · Communications Biology]

Morph-specific selection drives phenotypic divergence in color polymorphic tawny owls (*Strix aluco*) in Northern Europe

Corresponding Author: Dr Arianna Passarotto

Version 0:

Reviewer comments:

Reviewer #1

(Remarks to the Author)

This is an interesting paper investigating the processes of natural selection shaping plumage variation in an owl species with two discrete color morphs (tawny owl). The novelty of this research lays into the quantification and investigation of long-term temporal changes of continuous color variation within each morph, rather than focusing exclusively on changes in the frequency of each morph. The dataset compiled and used in these analyses is impressive, as it spans over 40 years and includes data on nearly 2000 measurements of coloration scores, which allows to draw robust conclusions. The manuscript is well written, although I still have some comments on the analyses and the presentation of the results.

L102-103: As far as I can judge from the model results in Table 1 this steadily increasing linear trend is non-significant or, let's say, only marginally significant ($p = 0.066$), which should be explicitly stated in the results.

L111: Parent-offspring regression is a fairly simple approach to estimate quantitative genetic variation in a trait and it can, actually, yield inflated estimates of heritability compared to approaches based on complex pedigrees and incorporating repeated measurements of a trait within individuals (see Åkesson et al. 2008. PLoS ONE 3:e1739). Hence, it would be highly recommended to support your results with some more advanced approaches, for example the so called animal model (Kruuk et al. 2000, PNAS 97:698-703). Also, some more details on parent-offspring regression are needed in the methods section, e.g. did you use midparent-midoffspring regression, where mean trait values were calculated across both parents and across different offspring?

L117: I am not sure whether recruitment success can be considered as breeding behavior, maybe the term "breeding performance" or "breeding traits" would better encompass your parameters.

L145-146: At the Fig. 4B the lowest color scores for grey morph actually coincide with high temperatures and few snow days, which is inconsistent with this statement.

L179-180: I understand "higher contribution" as being related to the effect size, but you cannot use F value as an effect size measure (it also depends on the degree of freedoms). Please either provide the true measure of effect size or rephrase/remove this statement.

L254: If this is due to migration and gene flow, it may be better to refer to "local phenotypic variation".

L280-283: This is only a speculation, as you did not test for this mechanism with your data, please tone down.

L287: Not particularly clear what is meant by the number of variant protein-coding gene alleles? Do you refer to any particular genes? Or allelic diversity across duplicated loci of any single gene? Needs clarification.

L308: ...persist or change in space and time?

L332: Why changes in quantitative traits should better reflect ecological than evolutionary shifts? I suppose that these shift

are often due to natural selection, so they reflect evolutionary (or microevolutionary) processes.

L352: If not being ringed before?

L378: Standard deviation is not a measure of repeatability, you should calculate and report intra-class correlation coefficients instead.

L383: Please report how many individuals were removed due to excessive variation in color scores.

L409: Do you mean both analyses? I can see only one approach here (GAM).

L414: It is stated in the methods that these trends were visually inspected, but it appears that they were also directly tested, as reported in lines 164-167.

L423: It is not entirely clear what this observer-morph combination refers to, please elaborate.

L436-438: How were environmental variables collected? On daily basis? Or as means across some periods (e.g. weekly?).

L438-439: It may be good to remind the reader that March 31st is the median laying date across years.

L440-442: Why recruitment success was scored as a binary trait? It seems that you lose a lot of biologically relevant variation with this kind of rough quantification. Was it unfeasible to use quantitative rather qualitative estimates of this parameter? Also, why recruitment success was quantified per pair of adults, instead of per individual? You quantified color scores on individual basis, so you also need individual data on recruitment success for your analyses.

L454: Do you mean recruits with both parental morphs known, or just a single parental morph?

L463: Should read "mean temperature and temperature variation"?

L419-466: It is redundant to repeat three times that "GAMs used thin plate regression splines for the smooth terms, 5 knots for the basis function, and restricted maximum likelihood (REML) for the smoothing parameter estimation", it will be OK to state it just once.

Fig.5: Why do the arrows point from laying date towards environmental variables (air temperature, snow depth and snow days)? Should it be the other way round, where winter weather affects laying date?

Reviewer #2

(Remarks to the Author)

In this study, authors use an exceptional dataset of colour variation in tawny owls collected over many decades to test how variation in colour phenotypes might be changing over time, and how genetics or environmental variables might be causing phenotypic variation. They look at these patterns in each morph separately as a way of exploring how divergence in morphs emerges.

I think that the question is neat, and the dataset is well suited to the study. In general, I found the manuscript to be well written and explained, with appropriate dealing of literature and inclusion of appropriate figures. I do, however, have several concerns about how the data were analysed and/or treated – which my comments below elaborate on. I also feel that there is a general tendency to over-reach in the setting up of the framework and also in the interpretation of results, and the findings are presented as if they are completely conclusive. I would urge authors to consider how they are presenting the theoretical framework of the study as well as the conclusions. Below are some general points, with links to places in the manuscript they refer to, that I hope the authors will find useful in revising their manuscript.

41 – 55; This first paragraph reads a bit like the authors are setting up a "straw-man" argument. I was quite shocked by the first few statements that claim that colour polymorphisms are insensitive to the environment and determined by few loci that follow Mendelian segregation. But then the authors appear to contradict those statements very quickly by then saying there is more continuous variation in coloration, which is polymorphic and also plastic (i.e., sensitive to the environment). In this general context to the manuscript should be edited.

Indeed, I think that in general, the framing of the paper in both the introduction and discussion could use some refining. I also think that some of the conclusions being drawn are either not supported by the statistics, or otherwise the statistics need adjusting.

60; it is not clear why authors cite climate change here

101; I see that for the brown morph there is a positive linear estimate, but in Table 1 the results are suggesting this is a non-significant result using an alpha of 0.05. How did authors determine if there was statistical support for effects throughout if not with an alpha of 0.05? I could not see explanation of that anywhere – apologies if I missed it. Looking at the data in Figure 2, it does look like there is essentially no change in the brown morph over time – which is what the model is telling us too – so I am curious about how the authors are interpreting this.

110; I couldn't find the explanation of how this model was fit in the methods

121 – 124; I don't really follow how these approximations of the rate of change are made from a smoothed term in the model – or if they are rather from the raw data. Figure 2 is not a great way of showing the temporal trend really as it's pretty hard to see the change in the response. Just from eyeballing those though, I could not see where these numbers came from.

I am also somewhat skeptical about the smoothed term being meaningful here as I think there is a risk of firstly overfitting the data – and secondly, cloudy interpretation of the effects from the model. I wonder if authors could provide more detail about where these numbers are coming from, or even better, consider fitting year as a linear and non-linear (e.g. quadratic) term in glms to better characterise the trend. Or else please justify and help the reader understand why gams were used.

129 & Figure 4; in the SEMs, I think that there should be an additional path that links year with colour score – that doesn't occur via either of the climate variables. I think this would be an appropriate addition to the analysis.

Section at 381; I think it's OK to treat the repeated measures by taking a median value, but it would also be possible to fit all the repeated measures directly in the model and include an id random term.

Statistical analyses starting at 401; it is not clear from any of the text whether authors fit all models assuming a gaussian distribution. I would imagine this to be the case in the absence of explaining otherwise, but I wonder if these data are appropriate for that assumption. Have the authors considered what distribution their data fit best? Also, there is some debate about whether one should fit year as an additional random effect in models trying to map temporal change in phenotypes. I have seen both ways, and I think it can be important to how we infer changes through time. I would suggest the authors consider doing so, or otherwise provide a thorough discussion and explanation of why they chose not to.

433; I was wondering why this time period was chosen for the environmental variables. I would think it would be ideal to be selecting the period in which the environment is assumed to be affecting the development of coloration – but finding that window I assume is tricky. Could authors provide some explanation of the choice of time period? One better approach to this is to run sliding window analyses, which would allow you to determine the window which affects the trait. I would also suggest that the authors re run analyses to decouple any changes in environment from trait change. Both might be changing, but some unmeasured variable might be causing that and the two are actually not linked. I think it's a bit risky to conclude anything about env-trait relationships at present.

Reviewer #3

(Remarks to the Author)

This manuscript uses a 43 year long time series data collected from a wild population of the tawny owl (*Strix aluco*) in Finland to assess to what extent color variation within two morphs (brown and grey) evolves differently over time and what are the drivers of phenotypic change. The authors assess 1) effects of morph, color score and key environmental parameters (temperature, snow) on laying date and recruitment success, 2) heritability of coloration (based on parent-offspring regressions), 3) morph specific selection on color score, laying date and recruitment success (using Structural equation, SEM, path models), and 4) potential genetic (paternal vs maternal morph) and plastic (environmental variables) on recruit colour score (using SEMs).

The authors find that the two morphs differ in patterns of color change over time (brown remaining rather stable, grey becoming lighter), heritability of color score is high (H2), that the morphs differ in predictors of color score (e.g. color score within the grey morph more strongly affected by temperature and snowy days) and recruitment success, as well as putative environmental effects in recruits (putatively stronger plasticity in the brown morph in response to temperature).

Overall, the study shows interesting differences between morphs in color variation and the putative predictors influencing these changes over contemporary time scales. The arguments are mostly well founded, the topic is broadly interesting for understanding intraspecific diversity of natural populations in general, and color variation in particular, the manuscript is quite clearly written and well-structured (though with some polishing needed), and the analyses seem sound (to the extent I am able to judge). My comments are therefore mostly editorial, as detailed below.

Specific comments

- It would be useful to explicitly state the generation time of *S. aluco* – as the time frame of evolution should be related to assumed number of generations in this system. A quick google from a non-expert suggests 1 year at first maturation which would imply (if true) the study lasted approximately 43 generations. However, it would be useful to provide explicit statements what is known about average generation time in the study population.
- The methods state that parent-offspring regressions were used to assess heritability, and figure 3A shows the regressions of mid-parent values with recruit values. However, the calculation of mid-parent values is not explicitly stated in the methods as far as I could see.
- The abstract seems a bit unpolished and could be slightly revised for smoother reading. For instance:

o L27 – I am not sure “color polymorphisms” should be called systems. Rather perhaps “Genetically discrete color polymorphic systems are used to study...”

o L32 – state clearly here that the system consists of two color morphs (grey and brown)

o L36 – you mean limited standing genetic variation?

o L37 – the relevance of continuous variation within discrete color morphs perhaps rather ?

- What is known about the extent and impact of color assortative mating in this system?

- The discussion is quite lengthy and could perhaps be shortened somewhat.

- The study was looking at predictors in winter/early life. Out of curiosity, is anything known about the differential effect of summer (which can vary in wetness and heat) on different colored individuals ? Are darker or lighter individuals able to cope differently with heat, for example?

- Snow depth is related to some of the variation, but it is unclear what snow depth selects for. Is it an indirect proxy for some agent of selection depicting winter conditions or does snow depth per se influence, for example, hunting success or diet availability ?

L48 – stating some of those evolutionary processes would be useful

L56-57 – I am not sure this is the right argument. First, are the morphs indeed stable over time? Why should we expect this? Perhaps you mean that “When color morph frequencies are stable over time, this implies a balance of selective forces acting on the population” ?

L64 – do you mean “continuous within morph variation but non-overlapping between morphs” here?

The last part of the introduction could be a bit better structured. Perhaps moving L87-89 to start of L71.

L97 onwards. It would help the reader to first shortly state the set up – something “Long-term monitoring data, from an area with nest boxes, of xx between years xx to xx”, rather than starting with the use of color scores. The description of the study system is quite rudimentary in the methods, as the system is described in earlier studies, so it would help the reader to grasp the context and set up better if some key statements were made in the result part.

L100 – do you mean divergence in plumage coloration of the morphs over time ?

L108 – the role of immigrant individuals, as well as “new cohort” should be more explicitly stated – what information do immigrant individuals convey? Presumably gene flow from elsewhere? And what are we to assess from the new cohort vs the rest of population biologically ? State these so reader is not left guessing.

L111 – State how many offspring are included in the analyses 170 pairs. State that this is broad sense heritability.

L118-119 – somewhat unclearly phrased what the proportion of new breeders was relative to immigrants

L124-126 – the argument seems somewhat circular

L143-146 – do grey individuals choose to breed in given conditions or is there potential, for example, for pre-selection through biased mortality or some other factor not implying active choice by the individuals ?

L161 – it is unclear what here is meant by “temporal shift between parents and offspring”. Rephrase for clarity.

L164 – you presumably mean smaller residuals over time – rephrase for clarity. To what extent are transgenerational effects possibly influencing the results?

L168 – by pedigree you mean 170 pairs? What is their relatedness? To what extent may relatedness (non-independence of the parent-offspring data points) influence the heritability estimates?

L169 – you state genetic factors also for colour scores, but presumably some component is due to plasticity ? I assume you refer to the high H^2 (0.83) which lends support for possible genetic variation within morphs also but still leaves the possibility of environmental and transgenerational effects?

L173-174 – to me the stronger maternal than paternal contribution also suggests there could be maternal effects. This should be stated as an alternative for sex-linked genetic mechanisms. Or if that is very unlikely then state so.

L191 – perhaps state here number of observations again

L192 – rephrase “indicates substantial within-morph variation in plumage coloration of *S. aluco*”

L195 – provide definition or reference for “pheomelanic” for benefit of non-experts

L197 – rephrase “...weaker plumage coloration across years” for clarity. Rephrase to “...contrasting changes within the two morphs in a highly heritable ...”

L199-200 – as well as presumably differences in the extent of plasticity

L201-202 – provide a reference

L203-204 – a bit of logic gap. What other life-history traits than ...? Previous part of sentence mentions no life-history traits.

L219-211 – reference for this statement missing

L226 – what is known about color variation in populations within dispersal distance of this system – or potential for color phenotype dependent dispersal ?

L230 - provide a reference for this statement

L236 – I don't quite follow the sentence of “which was also not the case in our population”. Rephrase.

L264 – somewhat unclear in the text what the developmental effects were – you presumably refer to results in Fig 5. Rephrase for clarity

L265-266 – Is anything known about the potential for maternal investment in egg content to have consequences for development and thereby color phenotype?

L270 – unclear to what result about early life-effects the text is referring to. Rephrase for clarity

L274 – you mean here color polymorphic systems to be mostly in butterflies and lizards? As other forms of polymorphisms are certainly studied in many other types of taxa (e.g. fish)

L292 – where is this result of 41% being due to additive genetic effects explicitly stated?

L305 – you refer here to colour polymorphism? In context of resource polymorphism the concept of individual specialization and its relationship to within species polymorphism is considered (see f.ex. Bolnick et al. 2003. The ecology of individuals: incidence and implications of individual specialization. *AmNat.*; Svanbäck & Persson 2004 “Individual diet specialization, niche width and population dynamics: implications for trophic polymorphisms” *J. Anim. Ecol.*)

L318-322 seems quite speculative. Also 322 – this seems an over statement from current study. While immigration and gene flow was suggested to be possibly important, it was not explicitly studied or its importance assessed as “important mechanism”. Rephrase (e.g. Which may be an important mechanism).

L330 – has the brown more spread across space such as across latitudine – or you mean it has become relatively more frequent over time?

L344 – the data was collected from two areas – and partially not overlapping in time. To what extent do these two subareas differ in color variation ? What was the number of samples in the current study from each site?

L347 – how long lived are the individuals? What was the mean and range of ages of the birds used in the study ? What is the age at maturation typically ?

L463 – is there a mistake here? Interactive term of temperature and temperature ? do you mean temp mean *temp CV (as in table 1, GAM6) - or you mean morph-temperature interaction?

L726 – explain midparent values in methods, as well as provide a reference for their calculation and short statement of potential short comings for estimating heritability.

L727 – I don't quite follow the wording here “dashed line denote transitions between morphs of parents of offspring” ?

L731-732 –The figure text seems to contain statements that are more like discussion – either move to discussion or provide references. The sentence about Mendelian inheritance requires a reference, likewise the potential for extra-pair paternity.

Version 1:

Reviewer comments:

Reviewer #1

(Remarks to the Author)

In my opinion the authors did a great job to revise the manuscript, I find it now all clear and sound. I have noticed one minor inconsistency in reporting the number of nestboxes (the authors report that each area was equipped with ca. 150 nestboxes, which does not sum up to the reported total of 360 nestboxes), but this can easily be fixed at the later stages of manuscript processing.

Reviewer #2

(Remarks to the Author)

I think that the authors have done a good job at dealing with most of the detailed comments they received, and I think that overall, the manuscript is reading much more clearly and concisely. I also appreciate that they have done some reanalysis in a few places. I do, however, still have a few issues in a couple of places about the statistics that I think have not been satisfactorily addressed or that I noticed on the authors adding new models.

First, I appreciate the authors response to my previous comment about the potential issue with using a median color score per individual. However, I don't think it has quite satisfied my concern. The authors correctly point out that the variation around each individuals median probably represent measurement error; but, then it makes more sense to me to propagate that error into their analyses by fitting repeated measures models. Otherwise, I think that authors need to consider how else to deal with measurement error, and/or acknowledge this in the manuscript.

Second, on re-evaluating the heritability analyses for which the authors have now included some animal model analyses (thanks for doing so – I think it's a nice addition), I was struck by how high this heritability is for all model types they implement. This may well be true – indeed the trait does appear rather simple genetically. However, I am concerned about the h^2 estimates being very inflated by the rather simple models that have been fit to the data which essentially only fit an intercept, additive genetic effects variance and residual term. Or, in the parent-offspring regression, it's a very simple model with no other terms but the parent trait value. In some simple laboratory systems, this would (perhaps) be appropriate. But, in this highly complex and large dataset from wild animals, I think this is almost certainly functioning to over-inflate the estimates of heritability. In particular, what I think is almost certainly happening is that heritability is being inflated because relatives share some component of their "environment", which at the moment will just look like high heritability. These factors could include, maternal environmental effects, temporal change caused by the environment (as indicated in other analyses), biotic or abiotic factors causing trait expression (i.e., plasticity) and so on. It seems to me that the authors have the data required to extend these models to get more accurate estimates of genetic effects, which are estimated once accounting for other/environmental causes of phenotypic variation. Incidentally, fitting the repeated measures dataset would allow you to fit a so-called permanent environment term which could help – but I would still encourage authors to consider how best to account for other environmental sources of phenotypic variation. See the following papers for more information and detail:

- Carys V Jones, Charlotte E Regan, Ella F Cole, Josh A Firth, Ben C Sheldon, Shared environmental similarity between relatives influences heritability of reproductive timing in wild great tits, *Evolution*, Volume 79, Issue 2, February 2025, Pages 220–231
- L. E. B. Kruuk, J. D. Hadfield, How to separate genetic and environmental causes of similarity between relatives, *Journal of Evolutionary Biology*, Volume 20, Issue 5, 1 September 2007
- Gervais, L., Morellet, N., David, I., Hewison, M., Réale, D., Goulard, M., Chaval, Y., Lourtet, B., Cargnelutti, B., Merlet, J., Quéméré, E. & Pujol, B. (2022). Quantifying heritability and estimating evolutionary potential in the wild when individuals that share genes also share environments. *Journal of Animal Ecology*, 91, 1239–1250

I see that in the SEM2, the authors are perhaps trying here to partition out environmental and genetic causes of phenotypic variation – but they are only rather specific environmental variables included, and I think there are probably quite a lot of unmeasured environmental factors that are affecting the trait – which should really be considered in both heritability and SEM analyses.

Due to one of the original reviewers being unavailable for re-review (Reviewer #3), this reviewer has also provided feedback on author response to points raised by Reviewer #3

regarding comments from Reviewer #3:

Mostly, I think that the authors have done an OK job at addressing the concerns of the previous reviewer, but I do think there is still room for improvement in some specific aspects of their concerns about modelling and interpretation of plasticity aspects of the manuscript. Specifically, I think authors still could improve on their in text (or even better in their analyses) justification of how they model plastic/environmental effects on plumage colouration. To me, this comes down to a couple of main points:

1. The authors are still using very specific time windows to test for environmental effects on the phenotype (both life history/fitness traits and plumage colouration). This isn't a problem per se - although really it would be much stronger if they expanded on this to quantitatively identify the important time windows - but there is very little justification of the choice or

acknowledgement of the limitations to this approach (e.g. omission of prenatal effects on plumage development etc. I think this is a nuanced and complicated issue because the time of year, or the period of development, where the environment can cause plasticity is really hard to have certainty on. I think the authors should be more careful in conclusions being drawn about environment-phenotype relationships as a result - unless they want to build on their analyses to help readers feel more confident.

2. The authors also take averages of environmental variables across a big time window for SEM1 (about 6 months) - which again needs some acknowledgment of this being a pretty coarse measure of how the environment might affect fitness.

3. I also see this reviewer pointed out the rather large heritability estimates, and flagged that some is certainly due to plasticity. I noted in my previous comments that these estimates of heritability come from models that do not account for plasticity, and I think those models should be updated so that they better partition variance between genetic and non genetic causes of phenotypic variation before we can be confident on heritability estimates.

With regard to the comments about maternal effects and transgenerational effects, this is a specific example relating to my above points. I think that the authors could reasonably expand their discussion to better incorporate other features like this that might be affecting phenotypic development - but that were not possible to include in the analyses here.

Version 2:

Reviewer comments:

Reviewer #2

(Remarks to the Author)

I appreciate the authors additions to the paper in response to my previous comments on earlier versions of this manuscript. I find the manuscript more appropriately cautious now, and I do think that the additional text they have added to justify various decisions and conclusions make it more accurate in its reporting. I think this manuscript will make a nice contribution to the literature. I only have one more comment that authors may wish to address textually: I couldn't find an explanation for how (with the associated justification) the genotype is coded in AM2 which models the additive and dominance effects. From checking the code and also Karell et al. 2011, I think authors are doing the same thing as in that paper. But – I think that a short explanation of how and why this is encoded as they have done would help the reader understand without having to follow the citation trail. Other than this, I congratulate the authors on a nice study.

-----**RESPONSES TO EDITOR(S) AND REFEREES**-----

Editor comments

In particular, please note that the following revisions would be necessary for us to contact our referees again:

- Revise statistical analysis to more rigorously account for repeated measures, using appropriate models as suggested by Reviewer 2.
- Provide justification for the chosen time period and consider revising analyses for environmental-trait relationships as suggested by Reviewer 2.
- Consider revising SEM based on suggestions from Reviewer 2.
- Reviewers also asked for more detailed methods: please note that we do not impose word limits and all methods should be described in enough detail so that they may be reproduced.

Response: *Thank you for your comments and guidance - as described in our cover letter, we have directly replied to each of the reviewer's comments below. We also*

Reviewer #1:

This is an interesting paper investigating the processes of natural selection shaping plumage variation in an owl species with two discrete color morphs (tawny owl). The novelty of this research lays into the quantification and investigation of long-term temporal changes of continuous color variation within each morph, rather than focusing exclusively on changes in the frequency of each morph. The dataset compiled and used in these analyses is impressive, as it spans over 40 years and includes data on nearly 2000 measurements of coloration scores, which allows to draw robust conclusions. The manuscript is well written, although I still have some comments on the analyses and the presentation of the results.

Response: *We are grateful for the positive assessment and all the insightful comments provided.*

L102-103: As far as I can judge from the model results in Table 1 this steadily increasing linear trend is non-significant or, let's say, only marginally significant ($p = 0.066$), which should be explicitly stated in the results.

Response: *Indeed, the brown morph has changed its plumage coloration only very little over the observed time frame. It was our oversight to not make this more explicit in the text (we previously indicated "marginally significant" relationships ($p=0.05-0.1$) in table 1 and discussed them in the text, but refrained from this in the version we submitted). Overall, this does not change our interpretation of the patterns (i.e., divergence is mostly driven by changes in the gray morph). We now mention this more explicitly in the text (LL 135-140) and in Table 1.*

L111: Parent-offspring regression is a fairly simple approach to estimate quantitative genetic variation in a trait and it can, actually, yield inflated estimates of heritability compared to approaches based on complex pedigrees and incorporating repeated measurements of a

trait within individuals (see Åkesson et al. 2008. PLoS ONE 3:e1739). Hence, it would be highly recommended to support your results with some more advanced approaches, for example the so called animal model (Kruuk et al. 2000, PNAS 97:698-703). Also, some more details on parent-offspring regression are needed in the methods section, e.g. did you use midparent-midoffspring regression, where mean trait values were calculated across both parents and across different offspring?

Response: *In the current study, we used a simpler approach because previous works from different populations of the same species yielded comparable estimates of heritability, which ranged between 72% (Brommer et al. 2005, Proc. R. Soc. B-Biol. Sci.), 79% (Karell et al. 2011, Nat. Commun.) and 93% (Gasparini et al. 2009, J. Anim. Ecol.). Our heritability estimate falls within this range of values, strongly suggesting that is consistent and unlikely to be (largely) overestimated (or underestimated). However, to back up those estimates we now include two animal models: one that corresponds to the model from Karell et al. (2011), and one that additionally accounts for additive and dominance effects of the genetic locus. Overall, these models support the results from the parent-offspring regression, indicating very high heritability (88%), of which a large portion seems to be explained through additive and dominance effects at the genetic locus. Please, see the new stand-alone paragraph in the results section (LL 152-167), text in the methods (LL 534-548) and newly added Table S3.*

L117: I am not sure whether recruitment success can be considered as breeding behavior, maybe the term “breeding performance” or “breeding traits” would better encompass your parameters.

Response: *We agree that the wording could be more precise and changed “breeding behavior” to “breeding performance” as suggested, both in the results (L 170) and in the methods (L 549).*

L145-146: At the Fig. 4B the lowest color scores for grey morph actually coincide with high temperatures and few snow days, which is inconsistent with this statement.

Response: *Yes, correct - we apologize for the mistake. We found it difficult to describe the non-linear pattern observed for the gray morph, and made another attempt at clarifying that passage (LL 194-205).*

L179-180: I understand “higher contribution” as being related to the effect size, but you cannot use F value as an effect size measure (it also depends on the degree of freedoms). Please either provide the true measure of effect size or rephrase/remove this statement.

Response: *We are sorry for that particular phrasing – the Reviewer is right of course. We meant that both the higher F-statistic for the main effect in the GAM and the larger effect size in the SEM of maternal than pattern color score on recruit color score might indicate a stronger maternal contribution to coloration. However, given the significant interactive effect in the GAM we should in any case be careful to interpret these main effects. We therefore removed this statement from the results and now interpret both statistical results in the discussion, albeit in a more toned-down fashion (LL 236-250).*

L254: If this is due to migration and gene flow, it may be better to refer to “local phenotypic variation”.

Response: *Our results do not allow us to unambiguously assess if the patterns are due to dispersal (this species does not migrate, but juveniles can travel fairly long distances during natal dispersal to find suitable territories to occupy) and gene flow. Still, we agree that what we observed in the study population might be a local process and we have rephrased as suggested (L 335).*

L280-283: This is only a speculation, as you did not test for this mechanism with your data, please tone down.

Response: *Agreed - we made substantial changes to that paragraph according to your and your colleague's comments (LL 370-391).*

L287: Not particularly clear what is meant by the number of variant protein-coding gene alleles? Do you refer to any particular genes? Or allelic diversity across duplicated loci of any single gene? Needs clarification.

Response: *We have now deeply restructured the paragraph and removed this statement (LL 370-391).*

L308: ...persist or change in space and time?

Response: *We see the Reviewer's point and have added “or change” (L 406).*

L332: Why changes in quantitative traits should better reflect ecological than evolutionary shifts? I suppose that these shift are often due to natural selection, so they reflect evolutionary (or microevolutionary) processes.

Response: *We agree with the Reviewer that, as phrased, our statement might be misleading. Here, we wanted to underscore that phenotypic changes over time, might not always reflect changes in the genetic makeup of populations, as we expect under evolutionary processes, but rather a rapid change in the interactions and dynamics within ecosystems. For instance, Ozgul and colleagues (2009, Science), found that the decline in body size of St. Kilda's sheep was primarily a consequence of environmental variation and not evolution. Although tightly entwined, ecological and evolutionary processes can contribute to phenotypic change quite differently. We have now rephrased this part to make it clearer, also removing some complexity (LL 432-442).*

L352: If not being ringed before?

Response: *Yes, we meant that. We have now clarified it (L 457).*

L378: Standard deviation is not a measure of repeatability, you should calculate and report intra-class correlation coefficients instead.

Response: *We used standard deviation to quantify the spread of multiple measurements around an individual's mean, reflecting the uncertainty in estimating that mean rather than repeatability per se. Our goal was to exclude individuals with the most uncertain phenotype estimates. While we refer to “repeatability” in the text, we acknowledge this is not a strict*

measure of it but used the term for its familiarity. We added a statement in the figure caption to clarify (LL 955-958).

L383: Please report how many individuals were removed due to excessive variation in color scores.

Response: *From the initial dataset of 1037 individuals with known identity (i.e., ring ID), we removed 19 individuals and their corresponding observations ($n = 55/2132$; see also Table S1). We have now explicitly reported this information in the text (L 489).*

L409: Do you mean both analyses? I can see only one approach here (GAM).

Response: *Correct. We substantially changed this paragraph and clarified by splitting GAM and SEM analyses. In addition, to address another comment, we have added also a new set of LM analyses to compare linear versus non-linear output (LL 508-533 and LL 550-554).*

L414: It is stated in the methods that these trends were visually inspected, but it appears that they were also directly tested, as reported in lines 164-167.

Response: *We agree with the Reviewer that a clarification is needed. We indeed tested for potential temporal changes in the heritability using the residuals, as we show in the text. We have now substantially restructured this part and add a stand-alone paragraph about the heritability estimate, as we run animal-model-based analyses (LL 551-555).*

L423: It is not entirely clear what this observer-morph combination refers to, please elaborate.

Response: *We were referring to potential biases stemming from three different observers, and how they score not only plumage coloration in general, but also in a particular morph. So instead of testing for an observer effect across both morphs, we tested for observer effects in a morph-specific fashion. The relevance of this approach is highlighted in Figure S2A (to which we refer in that sentence): the observer effect was much smaller for the brown morph than for the gray morph. We made an attempt to clarify this part (LL 521-523).*

L436-438: How were environmental variables collected? On daily basis? Or as means across some periods (e.g. weekly?).

Response: *The Finnish Meteorological Institute (FMI) gathers environmental variables on an hourly basis, which we have aggregated to a daily mean. We report this now in the text (LL 562-563).*

L438-439: It may be good to remind the reader that March 31st is the median laying date across years.

Response: *We have now slightly rephrased to include this information (LL 563-565).*

L440-442: Why recruitment success was scored as a binary trait? It seems that you lose a lot of biologically relevant variation with this kind of rough quantification. Was it unfeasible to use quantitative rather qualitative estimates of this parameter? Also, why recruitment success was quantified per pair of adults, instead of per individual? You quantified color

scores on individual basis, so you also need individual data on recruitment success for your analyses.

Response: *We see Reviewer's point, and we agree that using the number of recruits would provide finer biological information. Unfortunately, the recruitment rate in the Tawny Owl is particularly low (about 8% in the study population) due to high mortality in the post-fledging period and, usually, only single recruits per pair are recorded, which substantially reduces the variability in the number of recruits. Therefore, although coarse, we believe this classification reflects the biological information we are interested in, that is whether a pair actually produced successful individuals (at least one) that were recruited back into the breeding pool. Regarding the second point, we decided to consider recruitment success per breeding pairs and not per individuals because males and females share the breeding output, which could give rise to a pseudoreplication issue.*

L454: Do you mean recruits with both parental morphs known, or just a single parental morph?

Response: *We mean both parental morphs. We have now clarified it in the text (L 581).*

L463: Should read "mean temperature and temperature variation"?

Response: *We are sorry for the sloppy writing. Indeed, we meant "mean temperature and temperature variation". We have now corrected it (LL 590-591).*

L419-466: It is redundant to repeat three times that "GAMs used thin plate regression splines for the smooth terms, 5 knots for the basis function, and restricted maximum likelihood (REML) for the smoothing parameter estimation", it will be OK to state it just once.

Response: *We agree with the Reviewer. We have now moved this sentence to the general paragraph about the statistical analyses and removed it elsewhere (LL 511-513).*

Fig.5: Why do the arrows point from laying date towards environmental variables (air temperature, snow depth and snow days)? Should it be the other way round, where winter weather affects laying date?

Response: *The causal relationship you are describing is actually reflected in Figure 4A, where we test how winter conditions affect parental breeding behavior / laying date. The path analysis in Figure 5 focuses on the recruits, which could not influence their parents' timing of breeding. Hence, after the parents' decision to lay eggs, the offspring is exposed to the environmental conditions during the breeding period, which are the boxes after laying date (different from the boxes in Fig. 4, which are the environmental conditions of the entire winter period).*

Reviewer #2:

In this study, authors use an exceptional dataset of colour variation in tawny owls collected over many decades to test how variation in colour phenotypes might be changing over time, and how genetics or environmental variables might be causing phenotypic variation. They look at these patterns in each morph separately as a way of exploring how divergence in morphs emerges.

I think that the question is neat, and the dataset is well suited to the study. In general, I found the manuscript to be well written and explained, with appropriate dealing of literature and inclusion of appropriate figures. I do, however, have several concerns about how the data were analysed and/or treated – which my comments below elaborate on. I also feel that there is a general tendency to over-reach in the setting up of the framework and also in the interpretation of results, and the findings are presented as if they are completely conclusive. I would urge authors to consider how they are presenting the theoretical framework of the study as well as the conclusions. Below are some general points, with links to places in the manuscript they refer to, that I hope the authors will find useful in revising their manuscript.

Response: *We are grateful for the positive feedback and all the helpful comments.*

41 – 55; This first paragraph reads a bit like the authors are setting up a “straw-man” argument. I was quite shocked by the first few statements that claim that colour polymorphisms are insensitive to the environment and determined by few loci that follow Mendelian segregation. But then the authors appear to contradict those statements very quickly by then saying there is more continuous variation in coloration, which is polymorphic and also plastic (i.e., sensitive to the environment). In this general context to the manuscript should be edited.

Response: *It was not our intention to set up a straw-man argument, but our wording was indeed a bit unfortunate. We wanted to start off by making the point that, by definition, color polymorphisms result from direct genetic variation, where each color morph is clearly associable with a single genotype, which, however, can give rise to a range of phenotypic variation within a morph. This definition excludes ontogenetic color changes and polyphenism, which occurs when identical genotypes can produce different phenotypes depending on seasonal cycle (e.g. summer vs winter coats in stoats and snowshoe hares). By stating that the expression of color polymorphism is mostly or completely insensitive to the environment we meant that a color morph is neither susceptible to ontogenetic changes (i.e. age-mediated changes) nor reversible as for polyphenic forms. In summary, we agree with the Reviewer that, as structured, the statements may be misinterpreted. We have now made the text clearer (LL 56-75).*

Indeed, I think that in general, the framing of the paper in both the introduction and discussion could use some refining. I also think that some of the conclusions being drawn are either not supported by the statistics, or otherwise the statistics need adjusting.

Response: *We agree that some of the phrasing needed to be toned down, and we have tried to do so by addressing this and other Reviewers' specific comments.*

60; it is not clear why authors cite climate change here

Response: *We considered the following scenario: if a color polymorphism is stable in a specific population because each morph is adapted to a specific part of an environmental gradient (for example, temperature, Fig. 1A), then climate change may affect this stability by modifying the environmental gradient and associated ecological factors (Fig. 1B). In case of tawny owls, this means that gray morphs, which are considered to be better adapted to cold and snowy environments, will be maladapted under a warming climate, whereas brown*

morphs, which are considered to be better adapted to warmer environments, will benefit from climate change. Therefore, climate change - among other environmental changes - might affect the stability of polymorphism in the population, which is why we mentioned it here. We changed the sentence in an attempt to make this clearer (LL 80-83).

101; I see that for the brown morph there is a positive linear estimate, but in Table 1 the results are suggesting this is a non-significant result using an alpha of 0.05. How did authors determine if there was statistical support for effects throughout if not with an alpha of 0.05? I could not see explanation of that anywhere – apologies if I missed it. Looking at the data in Figure 2, it does look like there is essentially no change in the brown morph over time – which is what the model is telling us too – so I am curious about how the authors are interpreting this.

Response: *Our apologies - we previously indicated “marginally significant” relationships ($p=0.05-0.1$) in Table 1 and discussed them in the text, but refrained from this in the version we submitted. We corrected this in the text (LL 139-140) and in Table 1.*

110; I couldn't find the explanation of how this model was fit in the methods

Response: *We explain this in the statistics part of the methods (LL 525-528), but we have added some more context to make this more visible - please also note the changed model notation (GAM S1 / S2 -> GAM S1a / S1b).*

121 – 124; I don't really follow how these approximations of the rate of change are made from a smoothed term in the model – or if they are rather from the raw data. Figure 2 is not a great way of showing the temporal trend really as it's pretty hard to see the change in the response. Just from eyeballing those though, I could not see where these numbers came from. I am also somewhat skeptical about the smoothed term being meaningful here as I think there is a risk of firstly overfitting the data – and secondly, cloudy interpretation of the effects from the model. I wonder if authors could provide more detail about where these numbers are coming from, or even better, consider fitting year as a linear and non-linear (e.g. quadratic) term in glms to better characterise the trend. Or else please justify and help the reader understand why gams were used.

Response: *Given that the dynamics of owl color scores are inherently non-linear (GAM 1), we decided to also use GAMs to model laying date and breeding performance to be consistent, since these traits might also change in a non-linear fashion along with plumage coloration. The reported numbers are derived from a visual interpretation of the model predictions, which we obtained using the same number of knots (5) as in the color score models. We also carefully evaluated model performance (R^2 , QQ-plots, and residual distributions) to ensure the model was not overfit. However, we see your point that GAM-splines are neither as easy to interpret as linear fits, nor do they yield coefficients that can be used to quantify rates of change. We therefore decided to also fit a series of linear models to estimate these important traits in a linear fashion (LM 1-3; see LL 524-525 and LL 528-531). Additionally, we revised Figure 2 to make the corresponding panels larger, and to accommodate the visualization of both linear and non-linear fits. Finally, we report linear model statistics as well as the estimated coefficients in Table S2. We hope this clarifies this admittedly important part of our analysis.*

129 & Figure 4; in the SEMs, I think that there should be an additional path that links year with colour score – that doesn't occur via either of the climate variables. I think this would be an appropriate addition to the analysis.

Response: *Unfortunately, this was not possible - year is already strongly correlated with the climate variables, and adding a direct path to colour score would absorb nearly all of the available variance. We initially explored this option, and we found this to compromise the interpretability of the remaining pathways and lead to issues with multicollinearity and model convergence. We therefore modelled the effect of year indirectly via climate to maintain parsimony and model stability.*

Section at 381; I think it's OK to treat the repeated measures by taking a median value, but it would also be possible to fit all the repeated measures directly in the model and include an id random term.

Response: *Yes, we actually did this initially to confirm that the outcomes are the same, and then decided to use only the median value to i) explicitly treat the variation around the median as measurement error, and ii) keep the analytical approaches simple (linear models, GAMs, SEMs).*

Statistical analyses starting at 401; it is not clear from any of the text whether authors fit all models assuming a gaussian distribution. I would imagine this to be the case in the absence of explaining otherwise, but I wonder if these data are appropriate for that assumption. Have the authors considered what distribution their data fit best? Also, there is some debate about whether one should fit year as an additional random effect in models trying to map temporal change in phenotypes. I have seen both ways, and I think it can be important to how we infer changes through time. I would suggest the authors consider doing so, or otherwise provide a thorough discussion and explanation of why they chose not to.

Response: *Thank you, we forgot to mention which model families we used - we now state in LL 508-511 that we ran all models with a gaussian family, except in those with recruitment success as the dependent variable - there we used the binomial family. Regarding year as an additional random effect: in our case, year is a key predictor—we're directly interested in estimating changes over time. Modelling year as a random effect would absorb much of that variation, making it harder to interpret these trends. We therefore used smooth terms to capture nonlinear temporal patterns and account for autocorrelation in GAM 1. In the path analysis and GAMs 2-6, our setup is similar to a time-for-space substitution, leveraging variation among years to identify meaningful links between environmental change and population-level phenotypes, where among-year variation is less important.*

433; I was wondering why this time period was chosen for the environmental variables. I would think it would be ideal to be selecting the period in which the environment is assumed to be affecting the development of coloration – but finding that window I assume is tricky. Could authors provide some explanation of the choice of time period? One better approach to this is to run sliding window analyses, which would allow you to determine the window which affects the trait. I would also suggest that the authors re run analyses to decouple any changes in environment from trait change. Both might be changing, but some unmeasured

variable might be causing that and the two are actually not linked. I think it's a bit risky to conclude anything about env-trait relationships at present.

Response: *We analyzed winter climate variation using mean winter temperatures and the number of snow days, including all months with substantial snow cover (e.g., November–April), as harsh conditions in Scandinavia often extend beyond typical winter months. This broader window reflects periods that may affect adult survival and breeding timing, rather than the development of coloration, which is fixed in adults. This approach is consistent with prior studies in this system (e.g., Orlando et al. 2024 J. Ornithol.). For recruits, whose plumage develops during the nestling phase, we did focus on narrower, biologically relevant periods, specifically egg laying to fledging (Figure 4), where environmental conditions could directly influence color expression (e.g., via melanocyte activity during feather growth). Regarding sliding window analyses: we agree that they might be useful for identifying critical time windows, but this method is less compatible with our focus on long-term environmental trends and population-level phenotypic variation in adults, whose coloration is assumed stable. Our analyses aim to detect broader covariation between climate and phenotype, not immediate developmental effects. We agree that correlation does not imply causation, and both traits and environment may be influenced by unmeasured factors. Our goal, however, is to assess associative patterns over time, acknowledging these limitations, which is why we used a combination of coarser path analysis and more specific GAMs.*

Reviewer #3:

This manuscript uses a 43 year long time series data collected from a wild population of the tawny owl (*Strix aluco*) in Finland to assess to what extent color variation within two morphs (brown and grey) evolves differently over time and what are the drivers of phenotypic change. The authors assess 1) effects of morph, color score and key environmental parameters (temperature, snow) on laying date and recruitment success, 2) heritability of coloration (based on parent-offspring regressions), 3) morph specific selection on color score, laying date and recruitment success (using Structural equation, SEM, path models), and 4) potential genetic (paternal vs maternal morph) and plastic (environmental variables) on recruit colour score (using SEMs).

The authors find that the two morphs differ in patterns of color change over time (brown remaining rather stable, grey becoming lighter), heritability of color score is high (H^2), that the morphs differ in predictors of color score (e.g. color score within the grey morph more strongly affected by temperature and snowy days) and recruitment success, as well as putative environmental effects in recruits (putatively stronger plasticity in the brown morph in response to temperature).

Overall, the study shows interesting differences between morphs in color variation and the putative predictors influencing these changes over contemporary time scales. The arguments are mostly well founded, the topic is broadly interesting for understanding intraspecific diversity of natural populations in general, and color variation in particular, the manuscript is quite clearly written and well-structured (though with some polishing needed),

and the analyses seem sound (to the extent I am able to judge). My comments are therefore mostly editorial, as detailed below.

Response: *We are grateful for the positive assessment and for all the constructive comments.*

Specific comments

- It would be useful to explicitly state the generation time of *S. aluco* – as the time frame of evolution should be related to assumed number of generations in this system. A quick google from a non-expert suggests 1 year at first maturation which would imply (if true) the study lasted approximately 43 generations. However, it would be useful to provide explicit statements what is known about average generation time in the study population.

Response: *Tawny owls are long-lived birds that can live up to 20 years, but birds older than 7-10 years are uncommonly recorded in our population, where the average individual turnover is 4.46 years. Sexual maturity is indeed achieved when owls are 1 year old, although first breeding events usually take place when birds are 2–3-year-old, likely due to competition for territories during settlement phase after dispersal (e.g. Passarotto et al. 2022, Behav. Ecol.). Generation turnover is likely quite variable in this system due to, for instance, mortality in first year and variable skip breeding rate of territorial pairs due to unfavorable conditions (e.g. poor food availability). Although birds are actually able to reproduce after 1 year, we believe that adding a statement about number of generations could be misleading and would add too much complexity. Therefore, we would prefer not to add it.*

- The methods state that parent-offspring regressions were used to assess heritability, and figure 3A shows the regressions of mid-parent values with recruit values. However, the calculation of mid-parent values is not explicitly stated in the methods as far as I could see.

Response: *In the section “Statistical analyses” we explicitly wrote that we performed a linear model to estimate heritability of color scores and that we tested the residuals of the linear regression using an ordinary least squares regression with an autocorrelation term. We have now substantially changed this part as we ran an animal model to address a comment raised by Reviewer 2 (LL 531-548).*

The abstract seems a bit unpolished and could be slightly revised for smoother reading.

Response: *We agree – we did completely overhaul the abstract for better flow, and to add a bit more information. We tried to strike a balance between flow, comprehensiveness and length.*

L27 – I am not sure “color polymorphisms” should be called systems. Rather perhaps “Genetically discrete color polymorphic systems are used to study...”

Response: *We meant “study systems”, but this part got changed anyways when rewriting the abstract.*

L32 – state clearly here that the system consists of two color morphs (grey and brown)

Response: *We added this information as suggested (L 39). Please, be aware that we substantially changed the abstract (LL 35-50).*

L36 – you mean limited standing genetic variation?

Response: Correct – we added this in our new version of the abstract (L 43).

L37 – the relevance of continuous variation within discrete color morphs perhaps rather ?

Response: Indeed – this part got changed as well (LL48-50).

What is known about the extent and impact of color assortative mating in this system?

Response: In the study population, the sexes were not found to mate assortatively with respect to their color (see e.g. Brommer et al. 2005, Proc. R. Soc. B-Biol. Sci.). Therefore, we do not expect any bias in our results due to color assortative mating. We added a statement about this also in the methods where we present our new animal-model-based analyses for heritability estimate (LL 541-544).

- The discussion is quite lengthy and could perhaps be shortened somewhat.

Response: We agree with the Reviewer. We made an effort to streamline and trim the text.

- The study was looking at predictors in winter/early life. Out of curiosity, is anything known about the differential effect of summer (which can vary in wetness and heat) on different colored individuals ? Are darker or lighter individuals able to cope differently with heat, for example?

Response: This is a good point. We do not know whether and how summer conditions might impact morphs differently. We focused on winter conditions because they have been clearly associated with differential morph survival. In addition, winter conditions clearly influence breeding traits (e.g. laying date), while for summer conditions it may be more difficult to establish a link with fitness proxy. Considering that molecular analyses support the idea that the grey morph is the morph better adapted to cold (Baltazar-Soares et al. 2024 Mol. Ecol.), we might expect higher summer temperatures to be particularly detrimental for gray individuals. Similarly, it is not clear whether the brown morph might cope better with heat, although evidence shows that owl plumage redness increases in wetter and warmer areas (Passarotto et al. 2022 Glob. Ecol. Biogeogr.), suggesting a possible link between more pigmented colorations and physiological benefits under warmer (and more humid) environments. This aspect is certainly worth further research.

- Snow depth is related to some of the variation, but it is unclear what snow depth selects for. Is it an indirect proxy for some agent of selection depicting winter conditions or does snow depth per se influence, for example, hunting success or diet availability ?

Response: It might be a combination thereof. On the one hand, a thick snow cover is usually associated with very low winter temperature, and, therefore, snow depth could be a function of winter severity. On the other hand, snow cover could indeed directly affect tawny owls by reducing their hunting efficiency. Since prey population are also affected by climate in boreal areas, it is difficult to disentangle the selection operated by snow cover is direct or through prey availability or hunting success (see, for example, Orlando et al. 2024 J. Ornithol.).

L48 – stating some of those evolutionary processes would be useful

Response: *We have now added a few examples (L 63).*

L56-57 – I am not sure this is the right argument. First, are the morphs indeed stable over time? Why should we expect this? Perhaps you mean that “When color morph frequencies are stable over time, this implies a balance of selective forces acting on the population” ?

Response: *What we meant is that, theoretically, a color polymorphism can persist in a population only when morphs do not outcompete each other and this can be achieved when, on the long run, morphs accrue equivalent fitness. For example, a morph can show fitness advantages under certain circumstances and increase in frequency, but have a disadvantage when conditions change and therefore, its frequency will likely decrease. We agree that “frequencies” was not the appropriate term here and could generate confusion. We have changed it to “average fitness” (LL 77-78) also to better bridge this sentence with the following one.*

L64 – do you mean “continuous within morph variation but non-overlapping between morphs” here?

Response: *We meant that morphs may show additional continuous color variation within their own phenotypic spectra. We have now simplified the sentence to avoid confusion (LL 86-87).*

The last part of the introduction could be a bit better structured. Perhaps moving L87-89 to start of L71.

Response: *We have followed the Reviewer’s suggestion and moved this part to better structure the paragraph (LL 94-98). Moreover, we made the very last part of the introduction clearer with respect to the knowledge gaps and questions we are addressing with this system and our study (LL 108-111).*

L97 onwards. It would help the reader to first shortly state the set up – something “Long-term monitoring data, from an area with nest boxes, of xx between years xx to xx”, rather than starting with the use of color scores. The description of the study system is quite rudimentary in the methods, as the system is described in earlier studies, so it would help the reader to grasp the context and set up better if some key statements were made in the result part.

Response: *Here we respectfully disagree with the Reviewer: in the methods, we think we have given all the relevant information to understand the study population. We provide a concise but accurate description of the study area, including its extent and how it has been enlarged over time. However, we have added a brief sentence in the result section as suggested (LL 131-132) and, for sake of completeness, we have also added the total number of boxes monitored during the time period covered in this study (L 449).*

L100 – do you mean divergence in plumage coloration of the morphs over time ?

Response: *Correct, we were referring to the pattern we have found. We have now specified it in the text (L 135).*

L108 – the role of immigrant individuals, as well as “new cohort” should be more explicitly stated – what information do immigrant individuals convey? Presumably gene flow from elsewhere? And what are we to assess from the new cohort vs the rest of population biologically? State these so reader is not left guessing.

Response: *Correct, we contrasted immigrant vs resident owls to control for a possible effect of gene flow and to partly assess whether the pattern we observed is only restricted to our population or it might be more widely spatially spread and therefore not be a local phenomenon. We have now added a brief explicit statement (LL 145-146).*

L111 – State how many offspring are included in the analyses 170 pairs. State that this is broad sense heritability.

Response: *We meant 170 offspring (from as many pairs). We have now implemented new analyses about heritability using an animal model to address the comment of the Reviewer 1. We have therefore estimated the narrow sense heritability and substantially restructured this part in the results (LL 152-167) and in the methods (LL 534-548).*

L118-119 – somewhat unclear phrasing what the proportion of new breeders was relative to immigrants

Response: *Here, we mean that we only considered the recruits, which, by definition, are the individuals recruited back into the breeding population that are produced by known breeders. We just wanted to clearly highlight that any other new breeding individuals, like immigrants, are not counted. We have now tried to make the definition clearer (LL 171-172).*

L124-126 – the argument seems somewhat circular

Response: *We agree with the Reviewer that the sentence could be improved and we have now rephrased it (LL 178-180).*

L143-146 – do grey individuals choose to breed in given conditions or is there potential, for example, for pre-selection through biased mortality or some other factor not implying active choice by the individuals?

Response: *Here, we did not want to imply that grey individuals actively chose to breed under specific conditions as we are not able to say if this pattern is due to an active choice. On the contrary, since we did not find any linear relationship between coloration and laying date (i.e., breeding timing), we think that this pattern is probably due to a climate-driven selection on colored phenotypes either indirect (e.g., through pleiotropic effects) or direct, which could confer benefits like being better camouflaged under the main environmental condition (grey individuals would blend in with a snowy landscape more effectively than the brown ones; Koskenpato et al. 2020 Ecol. Evol.), increasing hunting success and, consequently, their breeding success.*

Morph-specific advantages under different circumstances might be of course reflected in differential survival (Karell et al. 2011 Nat. Commun.) and therefore variation in the breeding traits might just be a function of the individuals that survived the winter.

L161 – it is unclear what here is meant by “temporal shift between parents and offspring”.
Rephrase for clarity.

Response: *Here, we meant that we aimed to assess whether there was a tendency of the recruits over time to resemble less their parents. Please, be aware that we moved this part to a stand-alone paragraph (LL 152-167) and that we rephrased and simplified this particular sentence (L 162).*

L164 – you presumably mean smaller residuals over time – rephrase for clarity. To what extent are transgenerational effects possibly influencing the results?

Response: *We rephrased to improve clarity (LL 164-165). Moreover, while we acknowledge that transgenerational effects, such as parental or epigenetic influences, could contribute to the observed temporal decline in parent-offspring similarity, our analysis does not disentangle genetic inheritance from potential non-genetic parental effects, which may themselves shift across environmental gradients. To not make our discussion too speculative, we prefer to refrain from talking about transgenerational effects (although we now state that the potential non-genetic effects on color expression may indeed be transgenerational (L 350)).*

L168 – by pedigree you mean 170 pairs? What is their relatedness? To what extent may relatedness (non-independence of the parent-offspring data points) influence the heritability estimates?

Response: *The pedigree refers to 170 recruits for which we know both parents (indeed pairs) and have information about breeding period. Please, see also our response for L 111 above. We changed the phrasing after restructuring this part (L 155).*

L169 – you state genetic factors also for colour scores, but presumably some component is due to plasticity? I assume you refer to the high H^2 (0.83) which lends support for possible genetic variation within morphs also but still leaves the possibility of environmental and transgenerational effects?

Response: *Absolutely. This is what we try to emphasize by separating genetic from the environmental factors mentioned in the subsequent line (now L 228) and argument in the discussion. The genetic makeup of a population can be considered like a blueprint on which the environment, in sensu lato, can operate and determine variation upon, either through plasticity or epigenetics. In our study, 83% of (genetic) broad-sense heritability still leaves 17% of phenotypic variation unexplained, which is likely due to environmental and transgenerational effects. Although our data undermine the possibility to draw such conclusions, the environment might affect the expression of genes that regulate continuous coloration. We acknowledge that variation in parental color scores could also be part of the environmental factors affecting coloration if parental phenotypes affect parent performance, which in turn, might influence offspring phenotype. Thereby, similarity between the parents and their offspring might not be due (solely) to genetic effects. However, here we do not have a better variable to investigate the genetic component of color heritability and, considering that melanin-based coloration has a strong genetic basis (see e.g. San-Jose and Roulin 2017, Phil. Trans. R. Soc. B), the use of parental color scores to estimate the amount of variation in recruit coloration explained by genetic factors seems acceptable, although likely imprecise.*

L173-174 – to me the stronger maternal than paternal contribution also suggests there could be maternal effects. This should be stated as an alternative for sex-linked genetic mechanisms. Or if that is very unlikely then state so.

Response: *We agree with the Reviewer. We have restructured this section (LL 218-255) and added an alternative interpretation in the results (L 238) and in the discussion (LL 337-356).*

L191 – perhaps state here number of observations again

Response: *Done (L 258).*

L192 – rephrase “indicates substantial within-morph variation in plumage coloration of *S. aluco*”

Response: *Done. We have indicated the common name of the species as it might be more reader friendly (L 259).*

L195 – provide definition or reference for “pheomelanin” for benefit of non-experts

Response: *Done. We have now provided a definition in parentheses (L 262).*

L197 – rephrase “...weaker plumage coloration across years” for clarity. Rephrase to “...contrasting changes within the two morphs in a highly heritable ...”

Response: *We have clarified as suggested (L 265).*

L199-200 – as well as presumably differences in the extent of plasticity

Response: *We have now tried to integrate this possibility as well (L 268).*

L201-202 – provide a reference

Response: *We have now added two citations in support of our claim (L 270 and LL 675-679).*

L203-204 – a bit of logic gap. What other life-history traits than ...? Previous part of sentence mentions no life-history traits.

Response: *We agree with the Reviewer. We have now removed “other” (L 272).*

L219-211 – reference for this statement missing

Response: *Right. We are sorry for this oversight. We have now added the corresponding citation (L 294 and LL 682-684).*

L226 – what is known about color variation in populations within dispersal distance of this system – or potential for color phenotype dependent dispersal ?

Response: *We mention morph-specific dispersal for this study system in LL 332-334. Specifically, gray and brown owls were found to travel similar average natal dispersal distances, suggesting that natal dispersal is not affected by colour polymorphism per se. However, gray individuals do disperse further under colder winters, while brown ones show*

an opposite pattern and disperse further in year with milder winters. Indicating that is the interaction between temperature and coloration to likely affect tawny owl dispersal effort.

We don't know if variation in continuous coloration within the morphs may additionally explain some of the individual variation in dispersal distances, but previous studies have provided support for a relationship between the degree of melanisation, boldness and dispersal distances. Therefore, we cannot rule out this possibility.

Please, be aware that we have now restructured this part of paragraph in an attempt to shorten the text and removed this statement to keep it only in LL 419-423.

L230 - provide a reference for this statement

Response: *We have now added a proper citation (L 304 and LL 685-686).*

L236 – I don't quite follow the sentence of "which was also not the case in our population".
Rephrase.

Response: *We meant that also in our population, the increased mortality, with a consequent likely reduction of genetic variation, did not seem to affect the increase of the brown morph in the population. We have followed the suggestion and been more explicit to improve the clarity of our statement (LL 313-314).*

L264 – somewhat unclear in the text what the developmental effects were – you presumably refer to results in Fig 5. Rephrase for clarity

Response: *We meant the effects likely produced by the exposure to certain environmental conditions, both pre- or postnatally (since we consider the whole breeding period, from egg laying to fledging period), on the expression of coloration in recruits. We have rephrased the whole paragraph to make it clearer (LL 337-369).*

L265-266 – Is anything known about the potential for maternal investment in egg content to have consequences for development and thereby color phenotype?

Response: *In tawny owls, no study has investigated the effects of maternal investment in egg content so far. In other avian species, it is known that between-egg variation in female yolk production can translate into differences in offspring performance due to differences in lipid, protein or hormone content. Specifically for color phenotypes, an association between egg content and offspring coloration has been made in species displaying carotenoid-based colorations, where the pigments stored in the yolk seemed to affect the offspring coloration. We are not aware of any study dealing with differential egg investment and melanin-base plumage coloration, but it would certainly be worth further research.*

L270 – unclear to what result about early life-effects the text is referring to. Rephrase for clarity

Response: *We are sorry, but we could not understand what was unclear in this particular sentence. However, we have substantially changed the paragraph and this sentence is no longer present (LL 337-369). When we refer to early life conditions, we now specified we mean in the pre-fledging period (L 343) and, therefore, we refer to both pre- and post-natal environment (LL 357-358).*

L274 – you mean here color polymorphic systems to be mostly in butterflies and lizards? As other forms of polymorphisms are certainly studied in many other types of taxa (e.g. fish)

Response: *We absolutely agree with the Reviewer that color polymorphisms are investigated in a wide range of taxa. However, here we specifically refer to the study of morph-specific phenotypic plasticity and not to polymorphism as a whole. To the best of our knowledge, explicit tests for different plasticity between and within different morphs were carried out only in polymorphic species of lizards and butterflies. We have now slightly rephrased this sentence to avoid possible misinterpretations (LL 364-365).*

L292 – where is this result of 41% being due to additive genetic effects explicitly stated?

Response: *We have rephrased big portions of this part of the discussion as we run new analyses. We therefore removed this information (LL 388-390).*

L305 – you refer here to colour polymorphism? In context of resource polymorphism the concept of individual specialization and its relationship to within species polymorphism is considered (see f.ex. Bolnick et al. 2003. The ecology of individuals: incidence and implications of individual specialization. *AmNat.*; Svanbäck & Persson 2004 “Individual diet specialization, niche width and population dynamics: implications for trophic polymorphisms” *J. Anim. Ecol.*)

Response: *We refer to colour polymorphism, as stated in the precedent line, but we change “phenotypic” to “color” to avoid confusion (L 403).*

L318-322 seems quite speculative. Also 322 – this seems an over statement from current study. While immigration and gene flow was suggested to be possibly important, it was not explicitly studied or its importance assessed as “important mechanism”. Rephrase (e.g. Which may be an important mechanism).

Response: *We acknowledge that our study does not entirely support the statement. Here, we aimed to provide a possible scenario in the case that one of the morphs becomes limited in the expression of phenotypic variation and therefore in its capacity to trace fitness-related phenotypic optima. We have now rephrased this part and tried to make it less speculative (LL 415-423).*

L330 – has the brown more spread across space such as across latitudine – or you mean it has become relatively more frequent over time?

Response: *We were referring to the spread over time. We have rephrase this part of the text (LL 430-431).*

L344 – the data was collected from two areas – and partially not overlapping in time. To what extent do these two subareas differ in color variation ? What was the number of samples in the current study from each site?

Response: *We acknowledge that the areas are not overlapping in time. In Fig. S2 in the Supplementary material, where we controlled for the effect of different observers on any observed temporal patterns in tawny owl color scores, the green fit line identifies the single*

observer (“b”) for the sub-area added later in time (1987 onward) and, therefore, in a way, the area itself. It is possible to see that the patterns between the two areas differ but only in relation to the scoring of the observer, as the observer “b” tends to assign lower values. However, the trends over time in the two areas are comparable. In addition, given that the two areas almost overlap and individuals were recorded to move between them, we are confident that appreciable differences in color variation between the two areas are unlikely because they are not isolated.

Regarding the sample sizes, in the 43 years considered in this study, 783 breeding events were recorded for the first sub-area and 376 breeding events for the second sub-area added in 1987, summing a total of 1159 breeding events.

L347 – how long lived are the individuals? What was the mean and range of ages of the birds used in the study ? What is the age at maturation typically ?

Response: Please, see also our response for generation turnover at the beginning. Tawny owls can live up to 20 years. In our population, we have one record of a female breeding in the same nest box for 18 years. This is an exceptional record, as mean age in the population is 4.46 years. Sexual maturity is typically reached at 1 year, but the first breeding attempt starts when owls are 2/3-year-old, likely because of competition, as young owls probably stay in the area as floaters until suitable territories become vacant.

L463 – is there a mistake here? Interactive term of temperature and temperature ? do you mean temp mean *temp CV (as in table 1, GAM6) - or you mean morph-temperature interaction?

Response: We are sorry for the oversight. Indeed, we meant “mean temperature and temperature variation (i.e. CV)”. We have now corrected it (LL 590-591).

L726 – explain midparent values in methods, as well as provide a reference for their calculation and short statement of potential short comings for estimating heritability.

Response: We have added a brief explanation (L 846). Please, be aware that we have now calculated the narrow sense heritability using a simple animal-based model analysis; therefore, the corresponding paragraph has been rewritten (LL 152-167).

L727 – I don’t quite follow the wording here “dashed line denote transitions between morphs of parents of offspring” ?

Response: We have rephrased, changing “dashed lines” to “grid” and clarifying the sentence (LL 847-848).

L731-732 –The figure text seems to contain statements that are more like discussion – either move to discussion or provide references. The sentence about Mendelian inheritance requires a reference, likewise the potential for extra-pair paternity.

Response: We agree with the Reviewer and we removed these statements (LL 851-854). We partly moved this information to the Material and Methods section to better justify our heritability estimation (LL 542-543).

Reviewer #1 (Remarks to the Author):

In my opinion the authors did a great job to revise the manuscript, I find it now all clear and sound. I have noticed one minor inconsistency in reporting the number of nestboxes (the authors report that each area was equipped with ca. 150 nestboxes, which does not sum up to the reported total of 360 nestboxes), but this can easily be fixed at the later stages of manuscript processing.

Response: *We are grateful for the positive assessment and we are happy that the Reviewer found our changes satisfactory. We have fixed the inconsistency in the text (line 372).*

Reviewer #2 (Remarks to the Author):

I think that the authors have done a good job at dealing with most of the detailed comments they received, and I think that overall, the manuscript is reading much more clearly and concisely. I also appreciate that they have done some reanalysis in a few places. I do, however, still have a few issues in a couple of places about the statistics that I think have not been satisfactorily addressed or that I noticed on the authors adding new models.

Response: *Thank you for the positive feedback. We have given your concerns full consideration and tried to improve the analyses whenever possible. However, we found that we have limited statistical and analytical power for the heritability analyses, and we are unable to fit more specific models (e.g., maternal and common environmental effects in animal models). We now state these limitations more clearly in the text, explaining why we are not able to explore further the effects of certain variables on owl phenotype. Please, see our responses for each specific comment below.*

First, I appreciate the authors response to my previous comment about the potential issue with using a median color score per individual. However, I don't think it has quite satisfied my concern. The authors correctly point out that the variation around each individuals median probably represent measurement error; but, then it makes more sense to me to propagate that error into their analyses by fitting repeated measures models. Otherwise, I think that authors need to consider how else to deal with measurement error, and/or acknowledge this in the manuscript.

Response: *We appreciate the reviewer's attention to detail and agree that a repeated-measures structure would be appropriate if our goal were to model individual-level measurement error explicitly. However, our primary aim here was to estimate population-level changes in phenotypic composition, not within-individual dynamics. As such, repeated-measures models would introduce hundreds of biologically uninformative intercepts, increasing model complexity without improving inference. To address observer-related noise, we accounted for it directly in the model via an observer effect term (which revealed systematic differences), and addressed residual within-individual variability by replacing repeated scores with the individual's median value. This within-individual variation was small (Fig. S1) and not biologically meaningful, as individual plumage coloration in tawny owls does not change over time according to repeated color scoring (Brommer et al. 2005 and this study [Fig. S1]).*

Importantly, model results are virtually identical when using repeated measures versus median values (now stated in the manuscript, lines 452-454; also see figure attached below), confirming that our simplification does not affect inference. We now clarify this rationale in the Methods section and are confident that using denoised, median-based scores offers a statistically robust and biologically meaningful summary of individual phenotype for modeling population-level patterns.

- Brommer, J. E., Ahola, K. & Karstinen, T. The colour of fitness: plumage coloration and lifetime reproductive success in the tawny owl. *Proc. Biol. Sci.* 272, 935–940 (2005).

Second, on re-evaluating the heritability analyses for which the authors have now included some animal model analyses (thanks for doing so – I think it’s a nice addition), I was struck by how high this heritability is for all model types they implement. This may well be true – indeed the trait does appear rather simple genetically. However, I am concerned about the h^2 estimates being very inflated by the rather simple models that have been fit to the data which essentially only fit an intercept, additive genetic effects variance and residual term. Or, in the parent-offspring regression, it’s a very simple model with no other terms but the parent trait value. In some simple laboratory systems, this would (perhaps) be appropriate. But, in this highly complex and large dataset from wild animals, I think this is almost certainly

functioning to over-inflate the estimates of heritability. In particular, what I think is almost certainly happening is that heritability is being inflated because relatives share some component of their “environment”, which at the moment will just look like high heritability. These factors could include, maternal environmental effects, temporal change caused by the environment (as indicated in other analyses), biotic or abiotic factors causing trait expression (i.e., plasticity) and so on. It seems to me that the authors have the data required to extend these models to get more accurate estimates of genetic effects, which are estimated once accounting for other/environmental causes of phenotypic variation. Incidentally, fitting the repeated measures dataset would allow you to fit a so-called permanent environment term which could help – but I would still encourage authors to consider how best to account for other environmental sources of phenotypic variation. See the following papers for more information and detail:

- Carys V Jones, Charlotte E Regan, Ella F Cole, Josh A Firth, Ben C Sheldon, Shared environmental similarity between relatives influences heritability of reproductive timing in wild great tits, *Evolution*, Volume 79, Issue 2, February 2025, Pages 220–231
- L. E. B. Kruuk, J. D. Hadfield, How to separate genetic and environmental causes of similarity between relatives, *Journal of Evolutionary Biology*, Volume 20, Issue 5, 1 September 2007
- Gervais, L., Morellet, N., David, I., Hewison, M., Réale, D., Goulard, M., Chaval, Y., Lourtet, B., Cargnelutti, B., Merlet, J., Quéméré, E. & Pujol, B. (2022). Quantifying heritability and estimating evolutionary potential in the wild when individuals that share genes also share environments. *Journal of Animal Ecology*, 91, 1239–1250

I see that in the SEM2, the authors are perhaps trying here to partition out environmental and genetic causes of phenotypic variation – but they are only rather specific environmental variables included, and I think there are probably quite a lot of unmeasured environmental factors that are affecting the trait – which should really be considered in both heritability and SEM analyses.

Response: *We appreciate the Reviewer’s thoughtful comments regarding potential inflation of heritability estimates due to shared environmental effects. To address this, we extended our full animal model (AM2) to include key environmental predictors (temperature, temperature CV, snow cover, and snow depth) alongside additive and dominance effects of the candidate locus. This led to a modest increase in heritability from $h^2 = 0.12$ of AM2 to $h^2 = 0.17$, driven by a reduction in residual variance, suggesting that environmental variables account cumulatively for approximately 4% of the total phenotypic variation (lines 122-138).*

We did consult the references you provided and explored the possibility of adding maternal identity as a random effect to account for potential early-life influences. However, the vast majority of dams in our dataset have only a single recorded offspring (median = 1, mean = 1.62), providing insufficient within-dam replication to reliably estimate a maternal variance component. As a result, any models we tested that included a maternal term - both using frequentist and bayesian frameworks - were not identifiable and failed to converge. Unfortunately it seems what you suggest won't be possible with our dataset. However, following the literature you suggested, we did include in both AM1 and AM2 nest ID (i.e., the identity of the nest box where each recruit was born) as a random factor to account, at least partly, for common environment effects. We added similar statements about the limitations of

our data set in the Material and Methods section (lines 122-138), and are also more upfront in the Discussion about the implications for our heritability estimates (i.e., lines 287-288 and 309-312).

Reviewer #3 (Remarks to the Author):

Mostly, I think that the authors have done an OK job at addressing the concerns of the previous reviewer, but I do think there is still room for improvement in some specific aspects of their concerns about modelling and interpretation of plasticity aspects of the manuscript. Specifically, I think authors still could improve on their in text (or even better in their analyses) justification of how they model plastic/environmental effects on plumage colouration. To me, this comes down to a couple of main points:

Response: *Thanks again, also for taking over for Reviewer #3, we really appreciate your efforts.*

1. The authors are still using very specific time windows to test for environmental effects on the phenotype (both life history/fitness traits and plumage colouration). This isn't a problem per se - although really it would be much stronger if they expanded on this to quantitatively identify the important time windows - but there is very little justification of the choice or acknowledgement of the limitations to this approach (e.g. omission of prenatal effects on plumage development etc. I think this is a nuanced and complicated issue because the time of year, or the period of development, where the environment can cause plasticity is really hard to have certainty on. I think the authors should be more careful in conclusions being drawn about environment-phenotype relationships as a result - unless they want to build on their analyses to help readers feel more confident.

Response: *We acknowledge your concern regarding the use of specific time windows to model environment-phenotype relationships, and we agree that the temporal scope of environmental influence on developmental traits may have a big effect. However, we feel that our choice of timeframes - i.e., a half-year long window to reflect coarse between year winter dynamics, and a short post-egg-laying window to target the nest environment - was biologically justified. To address your concern we provide a sensitivity analysis that we now include as a supplementary figure (Figure S4), which we now explicitly mention in support of our decision for the given window sizes (line 166, lines 488-491).*

Specifically, we evaluated the effect of window size using cumulative averages of environmental conditions over increasing durations (200-7 days before egg-laying for adults, and 200-7 before and 7-100 days [to reduce window sizes smaller than 1 week] after egg laying for recruits), stratified by morph (Fig. S4). Across all four climate predictors, effect sizes remained small and largely non-significant (except snow days and air temperature for adults), indicating that our findings are robust to the exact length of the window used. We feel that this supports our decision to focus on a biologically meaningful two-month period, which overlaps with the average incubation time in our study population (~56 days) and also is consistent with evidence that maternal effects on plumage coloration are mediated during yolk formation and early development (Henriksen et al. 2011; Groothuis & Schwabl 2008).

- Grootuis, Ton G. G., and Hubert Schwabl. 2008. "Hormone-Mediated Maternal Effects in Birds: Mechanisms Matter but What Do We Know of Them?" *Philosophical Transactions of the Royal Society of London. Series B, Biological Sciences* 363 (1497): 1647–61.
- Henriksen, Rie, Sophie Rettenbacher, and Ton G. G. Grootuis. 2011. "Prenatal Stress in Birds: Pathways, Effects, Function and Perspectives." *Neuroscience and Biobehavioral Reviews* 35 (7): 1484–1501.

2. The authors also take averages of environmental variables across a big time window for SEM1 (about 6 months) - which again needs some acknowledgment of this being a pretty coarse measure of how the environment might affect fitness.

Response: *We agree that the six-month window used in SEM1 represents a relatively coarse measure of environmental conditions, but this was a deliberate choice to capture broad-scale interannual variation in winter severity (line 159). While some winter metrics (e.g. snow depth and snow days) are naturally correlated, our focus was not on fine-scale temporal resolution but rather on characterizing the overall intensity of winter across years (also see Fig. 2E-H). Finally, the lack of strong effects is perhaps not that surprising, given the strong heritability in this system (despite all difficulties to model it appropriately - see our response to your other comment).*

3. I also see this reviewer pointed out the rather large heritability estimates, and flagged that some is certainly due to plasticity. I noted in my previous comments that these estimates of heritability come from models that do not account for plasticity, and I think those models should be updated so that they better partition variance between genetic and non genetic causes of phenotypic variation before we can be confident on heritability estimates.

Response: *(Please also see our response to your comment above) To address this issue, we included a more complex animal model where, in addition to additive and dominance effects, we entered the same four key environmental variables to better control their impact on recruit phenotypes. Moreover, we added the identity of the nest as a random factor to account for possible common environment effects. According to this new model, the additive genetic effects have the largest impact on recruit phenotype, while the variance explained by nest ID was small but notable (8%). Among the four environmental predictors, only temperature variation (i.e. temperature CV) showed a positive, although marginally, significant trend, hinting at the possibility that higher temperature variation during the nesting phase may induce an increase in pigmentation and, therefore, providing preliminary support for plastic developmental changes. We have updated the passages that describe the model (lines 122-138) and discuss the outcomes (line 296-301).*

With regard to the comments about maternal effects and transgenerational effects, this is a specific example relating to my above points. I think that the authors could reasonably expand their discussion to better incorporate other features like this that might be affecting phenotypic development - but that were not possible to include in the analyses here.

Response: We now provide a specific example of transgenerational effects on the discussion (i.e., carry-over effects due to environmental variation and habitat quality) in line 287.